# Molecular Interaction of Genes Related to Anthocyanin, Lipid and Wax Biosynthesis in Apple Red-Fleshed Fruits

**DOI:** 10.3390/ijms262210987

**Published:** 2025-11-13

**Authors:** Sylwia Elżbieta Keller-Przybyłkowicz, Michał Oskiera, Agnieszka Walencik, Mariusz Lewandowski

**Affiliations:** 1Department of Horticultural Crop Breeding, The National Institute of Horticultural Research, Konstytucji 3 Maja 1/3, 96-100 Skierniewice, Poland; agnieszka.walencik@inhort.pl (A.W.); mariusz.lewandowski@inhort.pl (M.L.); 2Department of Microbiology and Rhizosphere, The National Institute of Horticultural Research, Konstytucji 3-go Maja 1/3, 96-100 Skierniewice, Poland; michal.oskiera@inhort.pl

**Keywords:** flavonoids, long-chain fatty acids, wax, cutin, *Malus domestica*, RNAseq, qPCR

## Abstract

Transcriptomic analysis of fruit flesh of the cultivars ‘Trinity’ (red-fleshed) and ‘Free Redstar’ (white-fleshed) uncovered a set of ten genes involved in different metabolic pathways. Three—*N3Dioxy*, *LAR1* and *F3Mo*—were mapped via phenylpropanoid and flavonoid biosynthesis (mdm00940, mdm00941); four—*AlcFARed*, *CER1*, *Cyp86A4* and *PalmTransf*—were mapped on the cutin, suberine and wax biosynthesis pathways (mdm00073); and three—*TropRed*, *CyP865B1* and *CytP450*—were mapped via the tropane/piperidine/pyridine alkaloid biosynthesis pathway and the peroxisome pathway (KEGG:mdm00960, KEGG:mdm04146). Our study highlighted the higher activity of *AlcFARed*, *CER1*, *PalmTransf* and *CYP86A4* in red-fleshed apple fruits and allowed us to discover a specific relationship between significant reductions in fruit wax coating and anthocyanin enrichment in fruit flesh. In addition, the uncovered inhibition of the *TropRed* gene and the activation of both *Cyp865B1* and *CYP86A4* suggests that both compounds generate primary alcohols and alkanes, ultimately bound to wax formation. Our results postulate that the fatty acid degradation process is initiated in the flesh of apple fruits and depends on the relationship between anthocyanin content and lipid and wax metabolism. These findings further our understanding of the molecular mechanism linking anthocyanin and wax, making it significantly important in the context of apple fruit storage stability.

## 1. Introduction

Apples are one of the most important fruit crops worldwide. Global annual apple production has reached over 80 million tons, and about 45% of production occurs in China, followed by Turkey and the USA [1]. The massive supply of this fruit to the market and its management raises challenges in terms of cultivation and storage. One way to address this situation involves increasing the consumption of fresh apples and their products (such as juices, chips and ciders), which can be achieved by introducing red-fleshed varieties into the apple industry [2].

In apples, selective breeding for red-fleshed fruit can be traced back to the 17th century discovery of *Malus pumila* var. *niedzwetzkyana* in forests in Turkestan, leading to development of red-fleshed cultivars with enhanced nutritional benefits [3,4,5,6,7,8].

One of the major health benefits of red-fleshed fruits is the presence of anthocyanins and other plant secondary metabolites, such as flavonoids, terpenoids, long-chain fatty acids and alkaloids. These elements have a huge impact on the development of apples and their physiological parameters and quality. They play an important role in coordinating the interactions between plants and the environment, as well as being involved in plant tissue protection [9,10,11,12].

Given anthocyanins’ health-promoting properties and increasing consumer demand for functional foods, apple breeding programs worldwide have intensified efforts to develop red-fleshed fruit varieties across multiple crops, including sweet orange, peach and kiwi [13,14]. Changes in breeding objectives, with a shift in desired apple characteristics from ornamental and processing features to functional fruit traits and fresh products, led to the cultivation of a series of red-fleshed apple cultivars, such as ‘jpp35’, ‘Weirouge’, ‘Baya Marisa’, ‘Redlove’ and ‘Meihong’ [15,16,17,18,19,20,21].

The anthocyanin biosynthesis pathway and the pigmentation of different tissues seems to be universal in the Plant Kingdom, having been clearly described for many species, such as strawberry [22], peach [23], blueberry, pomelo, citrus [24], tomato [25], *Ginkgo biloba* [26], etc. The precursors of and structural genes influencing anthocyanin biosynthesis such as phenlylalanine lyase (*PAL*), as well as chalcone synthase, isomerase (*CHS*, *CHI*), falvonon 3-hydroxylase (*F3H*), dihydroflavonol 4-reductase (*DFR*), anthocyanidin synthase (*ANS*) and UDP:flavonoid 3-O-glycosyltransferase (*UFGT*), have been clearly described [27].

Further investigation of the genetic control of pigmentation patterns in apple fruits underlined that the genetic basis of red-fleshed apple fruits is mainly related to the combination of the alleles of the MdMYB1 and mdMYB10 transcription factors [3,28,29,30].

Similar research uncovered other transcription factors (TFs) involved in anthocyanin pathway regulation, such as NAC, Ja2 and MADS [2,31,32]. TFs carrying structurally conserved DNA-binding domains, consisting of two repeats of R2R3, are strongly associated with anthocyanin biosynthesis. Further studies have also confirmed the dependence of specific MYBs on binding coregulatory proteins such as bHLH and WD40, forming the specific domain complexes required to activate the transcription of anthocyanidin structural genes, acting either early—EABG (*PAL*, *C4H*, *4CL*, *CHS*, *CHI* and *F3H*)—or late—LABG (*DFR*, *ANS* and *UFGT*)—in the biosynthesis pathway [28,33,34,35,36].

External environmental factors, including light and temperature, as well as internal environmental factors, such as sugar content and fatty acids, substantially affect anthocyanin biosynthesis, determining color development in both apple fruit skin and flesh [3,37,38].

Generally, sugars are important precursors in anthocyanin production, serving as signaling substances that can induce the cytoplasmic biosynthesis of this element [39,40]. They promote the glycosylation of unstable anthocyanidins by transforming UDP-Glc:flavonoid-3-O-glucosyltransferase (*UFGT*) to stable colorful (pink to purple in color) anthocyanins [41,42,43]. In *Arabidopsis*, sucrose and sucrose transporters (known as SUCs) can regulate the production of anthocyanin pigment factors (PAP), followed by the expression of the *DFR* (dihydroflavonol-4 reductase) structural gene, and finally induce anthocyanin synthesis and promote its accumulation [44,45,46,47,48]. It has also been observed that sucrose can activate *MdHXK1* gene-encoding hexokinase, which phosphorylates the MdbHLH3 transcription factor and mediates apple fruit coloration [49].

Despite efforts in apple breeding, red-fleshed varieties remain commercially limited due to the unfavorable acid–sugar balance and concerns about storage stability [50]. Regarding this issue, it must be underlined that the most important factors impacting apple fruit attractiveness are the intensity and composition of the cuticle (the waxy layer of the fruit’s skin).

The relationship between flavonoids and cutin formation was first described in tomato fruits by Heredia and coworkers [25]. They explained that some flavonoids are accumulated in cell vacuoles, and others are de novo synthesized and incorporated into the cuticle [25]. This indicates that the accumulation of anthocyanins results in reduced cutin formation, decreasing an apple’s potential shelf life [51].

Recent studies of the fruit skin of “Golden Delicious” (white-skinned fruits) and “Red Delicious” (dark red-skinned fruits) revealed varying levels of hydrolyzed cutin molecules and different compositions of aliphatic hydrocarbons, resulting in significant reductions in polysaccharides and phenolic compounds in red-skinned varieties [52]. This suggests a potential molecular link between pigmentation and fruit storage characteristics that remains poorly understood.

Very-long-chain fatty acids (VLCFAs) are pivotal in apple skin cuticle formation. In their biosynthesis, external metabolic pathways such as the tricarboxylic acid cycle (TCA) and glycolysis, which regulate carbon and acetyl CoA production, providing adequate precursors for fatty acid elongation, are employed [53,54,55,56,57,58]. Additionally, the cell wall invertase (CWI) gene plays a significant role in cuticle development, deposition and composition. Generally, CWI cleaves sucrose, irreversibly yielding glucose and fructose, which can be tracked via the hexose transporter. These hexose sugars are then taken up by plant cells via hexose transporters and may also act as signaling molecules in anthocyanin biosynthesis [59,60,61].

Recent research has confirmed the role of an additional mechanism in anthocyanin accumulation related to the transportation of primary synthesized anthocyanins from the pericarp to the cell vacuole, where they accumulate and generate the final fruit flesh color. Glutathione S-transferases (*GSTs*) [62,63], ATP-binding cassette (ABC-transporters) [64] and toxic compound extrusion metallothionein-transporters [65] seem to represent the key enzymes driving in this mechanism [66].

Since the fruit flesh color trait is very complex, current genetic models still inadequately explain the observed variations in apple flesh pigmentation intensity and its relationship with the fatty acid and wax biosynthesis pathways. This knowledge gap limits both breeding efficiency and our understanding of factors affecting fruit quality and shelf life.

In our study, we have uncovered new genes involved in alkaloid and general fatty acid biosynthesis, probably contributing to the final fruit wax coating process. Their expression profiles were validated in the fruit flesh of selected hybrid genotypes, derived from ‘Free Redstar’ and ‘Trinity’ crosspollination, which differ in fruit flesh coloration (white and red, respectively). This research may clarify the complexity of the anthocyanin accumulation and degradation processes, confirming the relatedness of anthocyanin formation and final cuticle disintegration in red-fleshed apple fruits.

## 2. Results

### 2.1. Significant Differences in Apple Genotypes with Regard to Total Anthocyanin Content

Measuring the spectral absorbance level in fruit juice with regard to pH points 1 and 4.5 allowed us to verify the phenotypical variance between fruits for the analyzed apple genotypes. The calculated average total anthocyanin content ranged from 9482 mg/100 mL (in ‘Free Redstar’, control) to 321 mg/100 mL (in genotype no. 48) (Table 1). The most informative and variable significance between evaluated samples is presented in Figure 1.

The calculated R square index (0.9920) explained the high variation between the individual apple genotypes tested. In addition, no significant (ns) correlation of variance between ‘Trinity’ and hybrid genotype number 154 was calculated (Table 2). As expected, the highest accumulation of anthocyanins was observed in seedling number 48, corresponding to its dark red fruit coloration of its flesh. Simultaneously, total anthocyanin content decreased when bright (pink, yellow to white) fruit flesh was present.

A summary of the statistical relationships, representing the correlation matrix between each of analyzed fruit samples, is presented in Table 2.

The data show that plant material significantly varies depending on the trait of interest and is properly characterized for validating the genetic backgrounds of newly uncovered differentially expressed genes. Furthermore, the individual fruit flesh samples from the above-selected genotypes were used for molecular analysis.

### 2.2. Comparative Analysis of Red and White Fruit Flesh Transcriptomes

Based on the RNA-seq experiment performed for ripe fruits of red-fleshed ‘Trinity’ and white-fleshed ‘Free Redstar’, we determined the difference between the transcript activities of candidate genes.

Four cDNA libraries were constructed and an average of 26,237,181 paired reads were obtained with the effective data volume for each sample, increasing to approximately 41,723,064 for ‘Free Redstar’ and 34,514,117 for ‘Trinity’. Clean reads were mapped on the apple reference genome (at a similar ratio,), with an average of approximately 89.7% unique mapping reads identified with proper pairing, giving a total of 54,299,271 mapped sequences (Appendix A).

Sample-to-sample cluster heat map analysis of expression profiles, as well as the total anthocyanin concentrations in the fruit flesh ‘Free Redstar’ and ‘Trinity’, indicated that the results were consistent, and the investigated cultivars samples were clearly grouped distinctly (Pearson correlation coefficient ratio: 0.8–0.9) (Figure 2a,b).

### 2.3. Identification of Functional Categories of DEGs

A comparative analysis of the RNAseq layouts of ‘Trinity’ and ‘Free Redstar’ allowed us to extract differentially expressed genes (DEGs) annotated with the *Malus* genome. The FPKM method was used to indicate the number of fragments per kilo base length of a protein-coding gene per million sequenced fragments.

Out of the 28,308 genes quantified (Appendix A), 7065 (25%) showed differential expression at *p* < 0.05, including 4866 (17.2%) at *p* < 0.01 and 3117 (11.0%) at *p* < 0.001. Among these, 1153 genes were up-regulated (FC ≥ 2) and 759 were down-regulated (FC ≤ −2); a further 1458 and 1434 genes exhibited moderate changes at 1 < FC < 2 and −2 < FC < −1, respectively (Figure 3). HISAT2 achieved 77.3–86.2% overall alignment, with 68.5–74.1% of read pairs aligning concordantly once and HTSeq assigning 54.5–57.7% of read pairs to gene features (Appendix A).

The functional categorization of DEGs allowed us to identify the two biggest groups of genes involved in general metabolism (1.065) and signal transduction (754). The other groups of genes were associated with transport (407), translation (378), transcription (309), stress responsive (308), protein modification (222), cell structure (194), hormone (107), development (30), cell division (29) and DNA repair (16). The most interesting groups of genes were involved in carbohydrate metabolism (75), lipid metabolism (57) and photosynthesis (44) (Appendix A). The top 50 DEG annotations are presented in Appendix A.

#### 2.3.1. Gene Ontology Enrichment of Differentially Expressed Genes (DEGs)

GO over-representation analysis of DEGs between ‘Free Redstar’ and ‘Trinity’ assigned functions across the three GO ontologies (Biological Process, BP; Cellular Component, CC; Molecular Function, MF). We detected extensive enrichment within BP (n = 3064 terms), CC (n = 488) and MF (n = 860) (Appendix A).

In BP, representative enriched terms included flavonoid and phenylpropanoid metabolism/biosynthesis (GO:0009812; GO:0009813; GO:0009698; GO:0009699), secondary metabolic processes and their biosynthesis (GO:0019748; GO:0044550), cuticle/cutin development (GO:0042335; GO:1901957), responses to Karrikin and toxic substances with cellular detoxification (GO:0080167; GO:0097237; GO:1990748; GO:0009636; GO:0098754; GO:0098869), macromolecule and protein methylation/alkylation/arginylation (GO:0043414; GO:0006479; GO:0008213; GO:0016598), the negative regulation of endopeptidase activity (GO:0010951) and the specification of stamen identity (GO:0010097) (Appendix A).

CC terms were enriched for ribosomal subunits and cytosolic ribosome (GO:0044391; GO:0022626; GO:0005840), the external encapsulating structure/cell wall (GO:0030312; GO:0005618) and megasporocyte and polar nuclei (GO:0043076; GO:0043078) (Appendix A).

In MF, enrichment was dominated by oxidoreductase/peroxidase activities (GO:0016491; GO:0004601; GO:0016684), acyltransferase activities (GO:0016746; GO:0016747), structural constituents of the ribosome (GO:0003735; GO:0005198), enzymes of the flavonoid pathway (naringenin-chalcone synthase and phenylalanine ammonia-lyase; GO:0016210; GO:0045548), antioxidant activity (GO:0016209), DNA AP endonuclease (GO:0003906) and abscisic-acid binding (GO:0010427) (Appendix A).

#### 2.3.2. KEGG Enrichment Analysis

Using gene identifiers obtained after mapping the transcriptome to the *Malus* reference genome (Appendix A), we performed KEGG pathway over-representation analysis (Appendix A; Figure 4). Among the top 60 enriched pathways, the largest gene counts/gene ratios were observed for mdm03010 (ribosome), mdm04014 (Ras signaling), mdm00940 (phenylpropanoid biosynthesis), mdm00941 (flavonoid biosynthesis), mdm00073 (cutin/suberin/wax biosynthesis), mdm02010 (ABC transporters), mdm00195 (photosynthesis), mdm05206 (MicroRNAs), mdm00750 (Vitamin B6 metabolism), mdm05010 (disease response), mdm04146 (peroxisome), mdm04141 (protein processing in endoplasmic reticulum) and mdm00960 (tropane piperidine/pyridine alkaloid biosynthesis).

For expression profile verification, ten representative genes were selected within key enriched pathways: (i) phenylpropanoid (confirming our validation efforts) and flavonoid biosynthesis (mdm00940, mdm00941; genes: *N3Dioxy*, *LAR1*, *F3Mo*; (Appendix A)); (ii) cutin/suberin/wax biosynthesis (mdm00073; genes: *AlcFARed*, *CER1*, *CYP86A4*, *PalmTransf*; (Appendix A)); (iii) tropane/piperidine/pyridine alkaloid biosynthesis and peroxisome (mdm00960, mdm04146; genes: *TropRed*, *CyP865B1*, *CytP450* (Appendix A)).

### 2.4. Activity of Selected Genes Depend on Anthocyanin Accumulation

Based on the comparative transcriptome analysis of red- and white-fleshed apple fruits, we verified the activity of a set of ten genes significantly influencing anthocyanin biosynthesis.

All selected genes were validated on cDNA samples from fruits produced via hybrid genotypes derived from the cross-breeding of ‘Trinity’ and ‘Free Redstar’, varying in accordance with the total anthocyanin concentration level (indicating fruit flesh color). Selected DEGs were grouped into clusters: genes involved in flavonoid biosynthesis, wax synthesis, peroxisome and tropane/piperidine and pyridine alkaloid biosynthesis.

#### 2.4.1. Expression Profiling of Genes Involved in Flavonoid Biosynthesis

The *N3Dioxy* gene (naringenin-3-oxydase, EC 1.14.11.9) was significantly over-expressed in the red-fleshed ‘Trinity’ variety. In general, the activity of this oxidase gene was relatively low in the studied fruits for all genotypes. Interestingly, lower activity (relatively to ‘Trinity’) in the dark red-fleshed fruits of genotype 84 was observed for this gene. Its highest activity was noted in the fruits of genotypes 141 and 154 (with pink- and light red-fleshed fruits, respectively) (Figure 5a). The *LAR1* gene (encoding leucoanthocyanidin reductase, E.C 1.17.1.3), which was selected for our study, also had significantly low activity in fruit flesh for all analyzed genotypes. As expected, since it participates in anthocyanin reduction, its activity was high in the reference samples of the white-fleshed fruits of the ‘Free Redstar’ cultivar (Figure 5b). Simultaneously, an inverse relationship was demonstrated for the *F3Mo* (flavonloid-3-monooxygenase, EC 1.14.13.21) gene, which was shown to be active in the red-fleshed fruits of the reference ‘Trinity’ cultivar and in the genomes of all hybrids producing fruit with red or pink flesh (Figure 5c).

#### 2.4.2. Expression Profiling of Genes Involved in Cutin and Wax Biosynthesis

The high activity of the selected genes, designated as involved in the synthesis of the wax and long-chain fatty acid pathways (*AlcFAred*—alcohol-forming fatty acyl-CoA reductase EC 1.2.1.84; *CER1*—very-long-chain aldehyde decarbonylase CER1, EC 4.1.99.5) and generally belonging to the FA reductases, was observed in samples of red-fleshed fruits (Figure 6a,b). For the gene encoding omega-hydroxypalmitate O-feruloyl transferase (*PalmTansf*, 2.3.1.188), negligible correlations were found between fruit flesh color, total anthocyanin content and the number of gene transcripts. However, interestingly, the number of transcripts of this gene was generally higher only in the fruit of the reference cultivar ‘Trinity’, as well as for hybrid genotypes 40 and 48, producing red-fleshed apples (Figure 6c). For the gene *CYP86A4* (recognized as FA carbonylase, CYP86A1), the highest expression was calculated for the red/pink-fleshed fruits of the genotypes 48, 40 and 44; however, the number of gene transcripts was lower in comparison to the red-fleshed fruits of the reference cultivar ‘Trinity’ (Figure 6d).

#### 2.4.3. Expression Profiling of Genes Involved in Tropane Piperidine and Pyridine Alkaloid Biosynthesis and Peroxisome

For the genes *CYP865B1* (flavonoid reductase activity) and *CytP450* (oxidase activity, EC 1.1.1.206), which were mapped via the peroxisome pathway, relatively higher expression levels, in comparison to the ‘Free Redstar’ cv., were observed in red-flesh fruits of ‘Trinity’ (Figure 7a,b). Similarly, higher activity for both genes was observed in fruit samples collected from the hybrid genotypes 103 and 77, producing yellow- or pink-fleshed fruits. For *TropRed* (tropinone reductase gene EC 1.1.1.206 (mapped through tropane/piperidine and pyridine alkaloid biosynthesis)), we observed a significant breakdown in its activity in all the tested fruit flesh samples (Figure 7c).

### 2.5. Validation of the Activity of Structural Genes from the Anthocyanin Biosynthesis Pathway in Accordance with Anthocyanin Accumulation

To verify the activity of the structural genes of the anthocyanin biosynthesis pathway (mdm00942) and confirm their undoubted relationship with the anthocyanin content evaluated in the analyzed fruit flesh samples, we calculated the number of transcripts of the *ANS* (anthocyanin synthase), *CHI* (chalcone isomerase) and *UFGT* (Flavonoid 3′-O-glucosyltransferase) genes. As expected, the relatively high activity of all structural genes was estimated for genotypes producing fruits with red or pink flesh.

Interestingly, the *ANS* expressions calculated for the hybrid genotypes 44, 40 and 84, producing red-fleshed fruits, was significantly higher compared to those for the red-fleshed fruits of the ‘Trinity’ cv. (Figure 8a–c).

## 3. Discussion

In the presented research, we have applied transcriptome analysis of ‘Trinity’ and ‘Free Redstar’ (which differ in accordance with apple fruit flesh color), which allowed us to uncover a new set of genes potentially related to flavonoid, wax (fatty acid degradation) and alkaloid biosynthesis.

In the first presented results, we have precisely verified that the plant material used for research purposes was chosen properly. Phenotypical analysis (with regard to TAC), as well as calculating the number of gene transcripts (for *ANS*, *CHI* and *UFGT*), confirmed high variation in the ten selected apple hybrid genotypes derived from cross-breeding the ‘Trinity’ and ‘Free Redstar’ varieties.

Through this pilot research, we have observed the significantly low activity of the *ANS*, *CHI* and *UFGT* structural genes (representatives of the anthocyanin biosynthesis pathway) in yellow- and white-fleshed fruits. The activity of the evaluated genes corresponded to lower total anthocyanin concentrations in the evaluated fruit samples. Similar results were described by Kondo and coworkers [66]. Based on this correspondence, the authors have highlighted the overexpression of *CHS*, *F3H* and *DFR*, followed by the down-regulation of *ANS* and *UFTG*, in the yellow/green skin of matured fruit of ‘Mutsu’ [67]. In general, this initial analysis confirmed the activity of the structural genes from the anthocyanin biosynthesis pathway, as well as the basic mechanisms of red-fleshed fruit, influencing the evaluated apple hybrid genotypes.

Interestedly, in our research, we have discovered ten genes potentially bridging the anthocyanin metabolism with other external pathways involved in the stimulation of their biosynthesis in apple fruit flesh. Those genes, mapped through the flavonoid biosynthesis, cutin/suberin/wax biosynthesis, tropane/piperidine/pyridine alkaloid biosynthesis and peroxisome pathways (Figure 9), have not been investigated so far.

### 3.1. Investigation of the Relationship of Uncovered Genes with Flavonoid and Pro-Anthocyanin Biosynthesis in Red-Fleshed Apple Fruits

Three genes—*N3Dioxy* (naringenin 3-dioxygenase), *LAR1* (leucoanthocyanidin reductase 1) and *F3Mo* (flavonoid 3-monooxygenase)—mapped through the flavonoid synthesis pathway (KEGG: mdm00941) were uncovered via transcriptome comparison of red-fleshed (‘Tinity’) and white-fleshed (‘Free Redstar’) apple cultivars.

The naringenin dioxygenase gene (*N3Dioxy*), uncovered in our study, was identified as playing an important role in the flavonoid biosynthesis pathway. This confirmed one of the workflow mechanisms of anthocyanin accumulation, followed by flavonoid incorporation in cuticles. This phenomenon, controlled by naringenin chalcone, together with flavonone 3-hydroxylase (*F3H*), was first described in tomato [25] and grapes [68]. In addition, we have discovered the relationship of *N3Dioxy* gen with the complex role of Flavonone 3-hydoxylase (*F3H*), which was previously intensively studied by [69]. Its activity highlights the promotion of flavonone synthesis in the condensation of malonyl-CoA with 4-coumaroyl-CoA, followed by the formation of naringenin chalcone, catalyzed by chalcone synthase (*CHS*), and subsequent conversion into naringenin by chalcone flavanone isomerase (*CHI*) [69]. Our findings underline the role of the *N3Dioxy* gene as a controller of flavanone biosynthesis in anthocyanin accumulation.

Moreover, we have confirmed that the flavonone 3′-monoxygenase gene (*F3Mo*, up-regulated in red-fleshed apple fruits) controls flavonoid biosynthesis by promoting naringenin dioxygenase gen (*N3Dioxy*) to achieve the final proanthocyanin formation, thus making it the compartment of the flavone biosynthesis complex (Figure 9). These insights had not been uncovered before this study.

Discovered in this study, *LAR*, which encodes leucocyanidine reductase and is assigned to the flavonoid biosynthesis pathway, is another regulatory gene, pivotal in the above-mentioned complex. After investigating the transcriptome of fruits of ‘Fumei’, Huo and coworkers confirmed the same role for the *LAR* gene in the flavonoid pathway [70]. The authors underlined that the *LAR* enzyme competes with the *ANS* gene in the conversion of leucoanthocyanidin into catechin [71,72]. This mechanism was also postulated by Liao and coworkers, who observed undetectable levels of *LAR* transcripts in the fruit skin of crab apples, which consisted mainly of catechin and low concentrations of polyphenols. Their observation underlined the role of *LAR* in suppressing late genes in the anthocyanin biosynthesis pathway, resulting in the loss of anthocyanin [73]. In contrast, Li and coworkers detected significant levels of *LAR1* and *LAR2* gene transcripts in the fruit skin of ‘Fuji’ [74]. This inverse relationship between *ANS* and *LAR* activity (up-regulation of *ANS* and inhibition of *LAR* in red- and pink-fleshed fruits, Figure 5b and Figure 8a) was also observed in our study (Figure 9).

### 3.2. Confirmation of Contribution of External Pathway Genes to Anthocyanin Biosynthesis

Since the relationship between flavanones/anthocyanins and cutin has not yet been reported in apples, in this research, we have confirmed the complex mechanism of anthocyanin accumulation, which requires outside metabolites to be activated.

In this study, four genes—*AlcFARed* (alcohol forming fatty acyl-CoA reductase), *CER1* (aldehyde decarbonylase), *PalmTransf* (omega-hydroxypalmitate O-feruloyl transferase) and *CYP86A1* (fatty acid hydrolase)—mapped via the cutin and waxy biosynthesis pathways (KEGG: mdm00073); two genes—*CYP865B1* and *CytP450* (probably flavoprotein reductase from cytochrome P450)—mapped through the peroxisome pathway (KEGG: mdm04146); and one gene—*TropRed* (tropione reductase)—mapped via tropane piperidine and pyridine alkaloid biosynthesis (KEGG: mdm00960) were found to be differentially expressed in the red-fleshed fruits of the cv. ‘Trinity’.

In our study, we have observed high activity of *AlcFARed*, *CER1*, *PalmTransf* and *CYP86A4* (recognized FA reductases, transformation of VLCFA, mapped in the cutin biosynthesis pathway (KEGG: mdm00073)) in red-fleshed apple fruits. These observations allowed us to uncover the specific relationships between significant reductions in the amount of VLCF released in fruits with higher anthocyanin contents.

While the apple cuticle is composed of very-long-chain fatty acids (VLCFAs, typically between C20–C34) derived from alkanes, alcohols, esters, aldehydes, ketones and triterpenoids [75,76], the role of these genes seems to be crucial. Those components are de novo synthesized in plastids via the catalysis of the fatty acid synthase complex (FAS) generally consisting of fatty acyl-ACP thioesterase (FAT) and fatty acid elongase (FAE). As stated by the authors, in contrast, long-chain fatty acids are disintegrated via acyl-reduction and decarboxylation pathways and mediated by fatty acid transferases [77] and fatty acid hydrolase (*CYP86A*) [78], as first recognized in this research.

Our observation confirm that triterpenoids are a group of molecules employed in cutin formation. They are derived from isopentenyl pyrophosphate (IPP, C5) and play a special role in plant cutin layer formation, proceeding with acetyl-CoA (energy precursor in different tissues) [61,79]. Other mechanisms are also applied in triterpenoid synthesis, such as squalene cyclization, hydroxylation, glycosylation and other structural modifications in which oxidosqualene cyclases (OSCs), cytochrome P450 monooxygenases (*CYP*) and glycosyltransferases (*UGTs*) are involved, which seem to be pivotal in cuticle formation [77]. Our data confirm these observations; thus, we have noted significant activation of *CYP* and *CER* genes in the fruits of the red-fleshed ‘Trinity’ cultivar (Figure 9).

The *CER1* gene revealed in our research was assigned to the group of *CER-like* genes previously discovered in different species such as *Arabidopsis* [80], tomato [81], sweet cherry [81] and orange [82]. Trivedi and coworkers have underlined their skin-specific expression, suggesting that these genes might be responsible for the differential accumulation of very-long-chain aliphatic compounds [51]; our data underlined the initiation of their activity in apple fruit flesh. As suggested, *CER* genes play an important role in alkane biosynthesis, being linked to the aldehyde biosynthesis process and the VLCFA decarbonylation pathway [82,83]. *FAR* is another type of gene involved in fatty acid transformation (such as omega-hydroxypalmitate O-feruloyl transferase) and reductases (such as alcohol forming fatty acyl-CoA reductase) and fatty acid hydrolases, generally hampering alcohol production, which is necessary for fatty acid elongation [61,77]. The *CER/FAR* interaction, which was first determined for *Arabidopsis*, suggests that both gene types generate primary alcohols and alkanes, which are finally associated with cuticular wax formation [84,85]. Our results confirm this relationship and highlight this mechanism for the first time in red-fleshed apple fruits.

One of the main factors impacting regulatory mechanisms in plant cytochromes and peroxisomes is light. Light may determine significant and intensive connections between those important cell organelles. Since light is essential for plant growth and development, some records underline their relationship with anthocyanin biosynthesis (negatively in apple fruits). Generally, activated photons inhibit the transcription factors [86]. This mechanism has not yet been investigated, and we have found some probable connections between anthocyanins and wax biosynthesis, as well as energy accumulation.

Since the final regulation mechanisms of phenylopropanoid biosynthesis are energy-intensive, there are several standalone oxidoreductases (*CYP*) responsible for their molecular transformation, catalyzed in cell structures such as cytochromes and peroxisomes, which take up the secondary metabolites derived from the aromatic amino acid phenylalanine in most plants [87].

As a result of these reactions, different fatty acid conjugates, plant hormones, secondary metabolites, lignins and other protective chemicals are produced [88,89,90].

In the membranes of the endoplasmic reticulum, electrons are transferred directly from NADPH to cytochromes via the NADPH-cytP450 reductase complex of flavoproteins, anchored to one layer of the membrane by a hydrophobic chain.

For some enzymatic molecules of cytochrome P450, such as cytochrome b5 and cytb5-cytP450 reductase, flavoproteins may participate in general electron transfer in the flavonoid cycle [91]. The *CytP450* gene belonging to the flavoprotein reductases discovered in our study seems to block flavonoid reductase, thus accelerating anthocyanin biosynthesis. This interaction in red-fleshed apple fruits has now been explained for the first time.

The crucial enzyme, which acts as a functional catalyst for the synthesis of intermediate 4-(1-methyl-2-pyrrolidinyl)-3-oxobutanoic acid, was finally transformed into tropinone through the catalytic activity of the cytochrome P450, which is a tropinone synthase (*CYP82M3*) [92], fully elucidated in belladonna flower petal pigmentation [93]. In our study, we have discovered a negative correlation between the down-regulated *TropRed* gene (mdm00960) and the up-regulated *Cyp865B1* and *CYP86A4* genes (mdm00073), leading to the activation of alkaloid biosynthesis necessary for final fatty acid elongation (Figure 9). For tropinone reductase (this enzyme has not been fully investigated, because it does not produce tropane alkaloids), we have observed negligible activity in evaluated apple red-fleshed fruits (Figure 9). This seems to be related to apple browning, caused by polyphenol oxidase (PPO) enzymes [94]. This mechanism was not observed in red-fleshed apples, underlining that red-fleshed fruits do not brown. However, this mechanism must be further investigated.

In this study, we have identified a new set of genes underlying the degree of relatedness between anthocyanin accumulation and cutin formation in the skin of ripe apple fruits.

Our results, for the first time, postulate that the fatty acid degradation process initiates in the flesh of apple fruits and depends on the relationship between anthocyanin content (which cover the color of the fruit’s flesh) and the activity of the genes regulating lipid and wax metabolism. This sheds new light on the mechanism involved, also accounting for apple storage stability. These findings suggest that red-fleshed fruits tolerate long-term storage conditions much less well than white-fleshed fruits.

Moreover, the genes identified in this study may provide a basis for developing functional molecular markers for flesh color corresponding to cuticle appearance in apples. They can also be used to monitor the studied traits and to select the most favorable new apple pre-breeding materials via marker-assisted selection (MAS).

## 4. Materials and Methods

### 4.1. Plant Material

For the RNA-seq experiment, ripe apple fruits of the red-fleshed ‘Trinity’ and white-fleshed ‘Free Redstar’ cultivars (minimum of 3 fruits per cv.) were collected from trees cultivated in an experimental orchard of the National Institute of Horticultural Research, Poland. Flesh disks (from peeled fruit), with a diameter of 2 cm and a depth of 1 cm, were dissected with a sterile blade. The samples were placed into liquid nitrogen and stored at −80 °C until RNA extraction.

The phenotypical assessment and validation of gene expression profiles was performed on a set of ten of the most perspective apple seedlings, derived via the cross-pollination of ‘Trinity’ and ‘Free Redstar’ cultivars (breeding program 2023). This set included the following genotypes: 40, 44, 48, 72, 77, 84, 103, 126, 141 and 154 (the visualization of fruit flesh color is presented in Appendix A). For the gene expression validation of ‘Free Redstar’, a control white-fleshed fruit (flesh pigmentation control) was used.

In total, 12 fruits collected from each hybrid genotype, as well as parental forms, representing different flesh pigmentation levels resulted from trait segregation and variable anthocyanin distribution in fruit flesh.

Two collected fruits per apple genotype were used for molecular analysis, and ten fruits per apple genotype were used for phenotypical characterization. All samples were immediately placed in liquid nitrogen and stored at −80 °C until analysis.

### 4.2. RNA Extraction

Total RNA from collected fruits (Section 4.1) was isolated, according to the manufacturer’s protocol, using the automated King Fisher Duo Prime station (Thermofisher Scientific, Life Technologies, Singapore) and the MaMax Plant RNA Isolation Kit (Applied Biosystems, Waltham, MA, USA) with magnetic binding beads (supplied with the kit). The removal of DNA residues in RNA preparates was performed with DNase I using the Turbo DNA-free kit (Thermo Fisher Scientific, Waltham, MA, USA). The concentration and integration of isolated RNA molecules were performed using a Bioanalyzer Agilent 2100 and Expert 2100 software (Agilent Technologies, Santa Clara, CA, USA).

### 4.3. Transcriptome Sequencing and Differential Expression Analysis

Transcriptome sequencing was performed by Genomed S.A. (Warsaw, Poland) using the Illumina NovaSeq 500 platform, generating 150 bp paired-end reads. Initial bioinformatic processing included quality assessment with FastQC v0.11.9 and adapter trimming using Cutadapt v3.4 (quality threshold ‘q = 25’; minimum read length ‘m = 15’). Reads shorter than 20 bp were discarded after trimming to ensure compatibility with downstream alignment. Reads were aligned to the *Malus domestica* ‘Golden Delicious’ reference genome assembly version ASM211411v1 (https://www.ncbi.nlm.nih.gov/data-hub/genome/GCF_002114115.1/ accessed on 6 April 2025) using HISAT2 v2.2.1 with the ‘--rna-strandness RF’ option for stranded data. Gene-level quantification was performed with HTSeq using the ‘--stranded = reverse’ parameter. Differential expression analysis comparing the ‘Free Redstar’ and ‘Trinity’ cultivars was performed using DESeq2 v1.38.3 in R v4.2.1. Genes with a *p* < 0.05 and absolute log_2_ fold change > 1 were considered to be differentially expressed. Downstream analyses, including quality control, DEG identification and visualization with EnhancedVolcano v1.14.0, were implemented through custom R scripts.

### 4.4. Gene Annotation, GO and KEGG Enrichment Analyses

Functional annotation and gene model refinement were performed using AGAT v0.5.1 for GTF/GFF processing and unique feature ID assignment, and gffread v0.12 was used for file conversion and sequence extraction. Transcript-to-gene mappings were generated using the R package txdbmaker (Bioconductor v. 3.22) from GTF files. Protein sequences extracted from the genome annotations were screened and functionally annotated using multiple resources: NCBI (gene_info, gene2go files downloaded 1 July 2025), InterProScan v5.59-91.0 for domain identification and Gene Ontology (GO) term assignment, Pfam-A database (via hmmscan) for protein family detection and EggNOG mapper with the Viridiplantae taxonomic scope for orthology and functional annotation. KEGG pathway analysis was performed using KEGGREST for identifier mapping and Pathview v1.48.0 for pathway visualization incorporating log_2_ fold change information. The annotations used merged GO term sources and functional keywords across description, KEGG_Pathway and GO_terms columns to classify genes into major biological categories, including metabolism, signaling, development, DNA repair and stress response.

All downstream enrichment, visualization and figure production stages were carried out in R v4.2.1 utilizing clusterProfiler v4.16.0, GO.db v3.21.0, enrichplot v1.28.4, EnhancedVolcano, KEGGREST v1.48.1, pathview v1.48.0, ggplot2 v3.4.0, p-heatmap and ComplexHeatmap v2.12.1 to ensure reproducibility and publication quality.

Raw sequence data are available in the NCBI Sequence Read Archive (SRA) under BioProject accession PRJNA1339032.

### 4.5. cDNA Synthesis and qPCR Tests

Three independent fruits from each selected apple seedling were collected for total RNA (1 µg) isolation, as described in Section 4.2. The concentration and integration of isolated RNA molecules was calculated using Bioanalyzer Agilent 2100 and Expert 2100 software (Agilent Technologies, Santa Clara, CA, USA); then, the obtained RNA preparations were reverse-transcribed into cDNA using the AffinityScript QPCR cDNASynthesis Kit (Agilent, Santa Clara, CA, USA). The reverse transcription reaction, with the universal oligo-dT primer and reverse transcriptase (RT), was carried out under optimized thermal conditions, i.e., 25 °C for 5 min, 42 °C for 5 min (oligo-dT annealing), 55 °C for 15 min (reverse transcription) and 95 °C for 5 min (enzyme inactivation), using a Biometra Basic thermocycler (Biometra, Göttingen, Germany). The expression profiles of selected differentially expressed genes were estimated via qPCR tests performed with specific oligonucleotides designed within the study (Primer3plus software (https://www.bioinformatics.nl/cgi-bin/primer3plus/primer3plus.cgi, accessed on 20 June 2024)). Gene-encoding *ACTIN* [95] was used as a qPCR data normalizer (Table 3).

QPCRs were performed using SYBR Green fluorescent dye (Kapa SYBR qPCR kit, KapaBiosystems, San Francisco, CA, USA) through a RotorGen 6000 thermal cycler (Corbett Research, Sydney, Australia). Two pairs of specific primers, complementary to the selected target DEGs, were used in analogous reactions. The cDNA template was prepared in dilutions of the estimated concentrations, enabling the preparation of a standard amplification reaction curve. The thermal profile of a single qPCR was as follows: 95 °C for 5 min (polymerase activation), followed 40 cycles including the following steps: 95 °C for 15 s (denaturation), 60 °C for 20 s (oligonucleotide annealing) and 72 °C for 20 s (fluorescence level detection). Relative expression (fold change), normalized with regard to *ACTIN*, was determined using Rotor-Gene 6000 Series Software 1.7 (Corbett Research, Sydney, Australia) based on single-data-point amplification curve threshold cycles (Ct values) (2^−∆∆Ct^) [96]. The average value of relative expression was normalized to the white-fleshed fruit control ‘Free Redstar’ cultivar. The standard errors of the mean ± SEM and t-significance at *p* < 0.05, *p* < 0.01, *p* < 0.001 and *p* < 0.0001 between the ‘Free Redstar’ and red-fleshed cultivars were calculated separately (GraphPad Prism 10.0.3, Dotmatics, Boston, MA, USA). Relative fold change diagrams for each gene were drawn using GraphPad Prism 10.0.3.

### 4.6. Anthocyanin Measurements and Fruit Phenotypical Assessment

Ten peeled fruit samples (5 g of fresh mass) for each selected apple genotype (which differ in accordance with flesh coloration) were collected for total anthocyanin concentration measurements. For this purpose, the pH differential method [97,98] was used.

The fresh samples ground in liquid nitrogen were moisturized with 20 mL of pH 1.0 (25 mM KCl and 0.1 M HCl) and pH 4.5 (0.4 M CH_3_COONa) buffers, incubated for 20 min at room temperature and then centrifuged (7000 rpm) at 4 °C for 15 min. The absorbance of the fractionated supernatant was read at 520 and 700 nm using the spectrophotometer VIS V500 (BioSens, Warsaw, Poland). The total anthocyanin concentration (TAC) was calculated via the following formula: TAC $=\frac{A\;V}{M}$. Here, A = (A520 nm − A700 nm), with pH 1.0 (A520 nm–A700 nm) to pH 4.5; V = volume of extract (mL, total volume 25 mL); and M = mass of the sample (g). The calculation of anthocyanin concentrations was based on a cyanidin-3-glucoside standard solution (molar extinction coefficient = 25.965 cm^−1^; molecular weight = 449.2 g mol^−1^). The results were expressed as mg of cyaniding-3-glucoside equivalents per 100 mL of sample supernatant.

### 4.7. Statistical Analysis

Phenotype–genotype associations between anthocyanin content and candidate gene expression were computed via GraphPad Prism v10.0.3 using Pearson’s r. Differences in anthocyanin content were tested through two-way ANOVA with appropriate post hoc comparisons. Simple linear regression summarized dose–response trends where applicable.

Differential expression. HTSeq gene counts were analyzed using DESeq2 v1.38.3 (the Wald test). The testing universe comprised all genes retained after independent filtering. DEGs were defined as raw *p* < 0.05 with |log_2_FC| > 1. Quality control included PCA, MA and volcano plots.

Functional enrichment (ORA). clusterProfiler was used for GO (BP/CC/MF) and KEGG over-representation with the DESeq2 testing universe. Enrichment was performed at raw *p* < 0.05 (clusterProfiler pAdjustMethod = “none”). GO resources were integrated from GO.db and InterPro/Pfam; KEGG IDs were mapped using KEGGREST, and pathways were visualized using Pathview (log_2_FC overlays).

*GSEA*. Rank-based enrichment (gseGO, gseKEGG) used genes ranked by the DESeq2 Wald statistic; significance was set at a nominal *p* < 0.05 (default permutations), with normalized enrichment scores reported.

All analyses were performed using R v4.2.1. Figures were generated using ggplot2 v3.4.0, ComplexHeatmap v2.12.1, pheatmap and EnhancedVolcano v1.14.0. Unless stated otherwise, tests were two-sided and statistical significance was assessed at nominal *p* < 0.05 (unadjusted).

## 5. Conclusions

Our analysis revealed novel molecular interactions between anthocyanin biosynthesis and wax metabolism pathways in apple fruit flesh. The identification of ten key genes—*N3Dioxy*, *LAR1* and *F3Mo* (flavonoid pathway); *AlcFARed*, *CER1*, *PalmTransf* and *CYP86A4* (wax biosynthesis); and *TropRed*, *CyP865B1* and *CytP450* (alkaloid biosynthesis/peroxisome)—provides new insights into the molecular basis of flesh coloration and storage stability relationships.

The identified genes have immediate potential as molecular markers for marker-assisted selection (MAS) in red-fleshed apple breeding programs. Priority should be given to developing functional markers for *CER1* and *AlcFARed* expression levels as predictors of storage stability, as well as the *F3Mo/LAR1* ratio as an indicator of flesh color intensity. A combined marker assay should be developed for the simultaneous selection of color and storage traits.

Our findings suggest that red-fleshed varieties’ storage limitations stem from their reduced wax biosynthesis capacity. Future research should focus on developing cultivation practices to enhance *CER1* and *CYP86A4* expression, investigating postharvest treatments targeting wax metabolism pathways and exploring genetic modification approaches to restore wax biosynthesis in fruits with high-anthocyanin backgrounds.

Successfully addressing storage limitations could unlock significant market potential for red-fleshed apples within functional food sectors. Improved varieties would support sustainable agriculture by providing growers with premium products while delivering enhanced nutritional benefits to consumers.

Future investigations should validate these gene–trait relationships across diverse genetic backgrounds, develop high-throughput screening methods for breeding programs and explore environmental factors influencing the anthocyanin–wax metabolism balance. Integration with genomics-assisted breeding platforms will be essential for applying these molecular insights to commercial cultivar development.

## Figures and Tables

**Figure 1 ijms-26-10987-f001:**
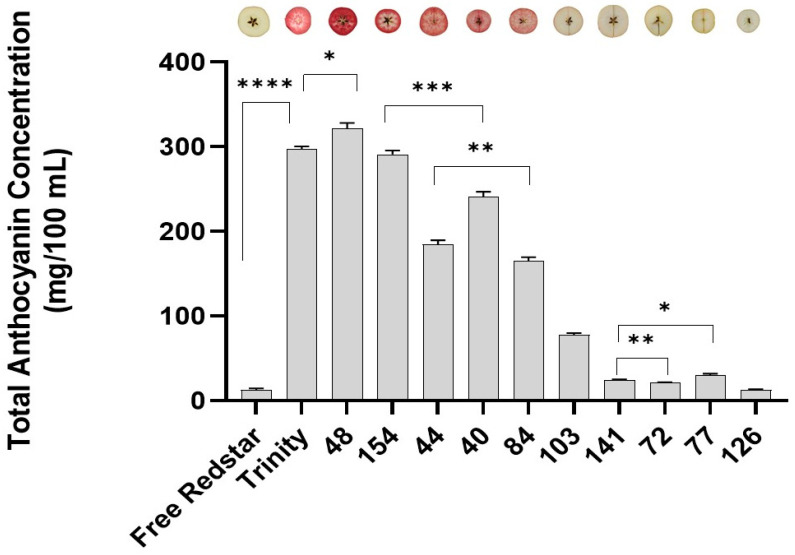
The total anthocyanin concentrations in the apple fruit flesh of parental and hybrid genotypes, evaluated spectrophotometrically and compared with regard to cyanidin-3-glucoside. The data are presented as the average anthocyanin concentrations with the standard error of the fruit mean (±SEM) compared to ‘Free Redstar’ white-fleshed fruit c. The *t*-test had significance levels of * *p* < 0.05, ** *p* < 0.01, *** *p* < 0.001 and **** *p* < 0.0001, recorded using GraphPad Prism 10.3.

**Figure 2 ijms-26-10987-f002:**
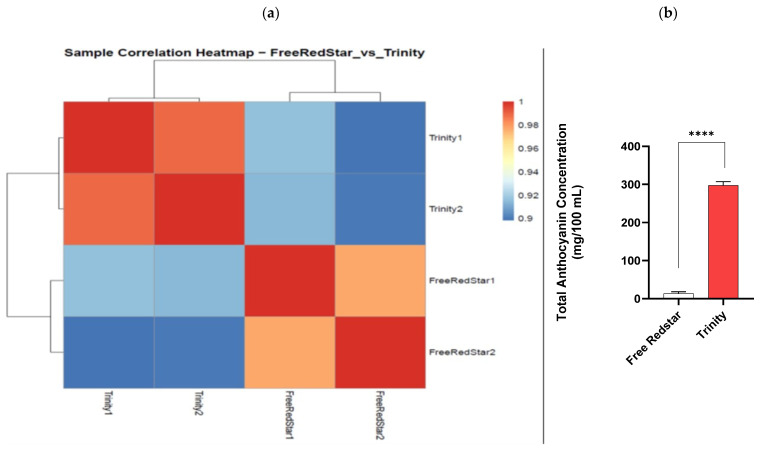
(**a**) Heat map correlation between ‘Free Redstar’ and ‘Trinity’; (**b**) phenotypical characteristics of total anthocyanin concentrations in white-fleshed ‘Free Redstar’ and red-fleshed ‘Trinity’ (**** *p* < 0.0001).

**Figure 3 ijms-26-10987-f003:**
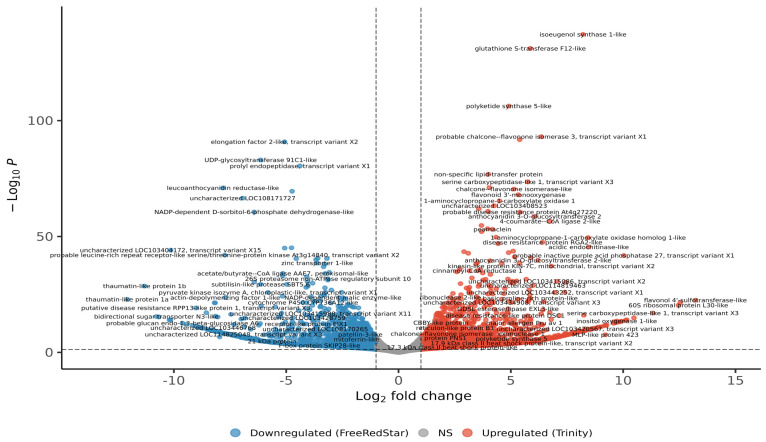
Volcano plot distribution of annotated up-regulated and down-regulated DEGs.

**Figure 4 ijms-26-10987-f004:**
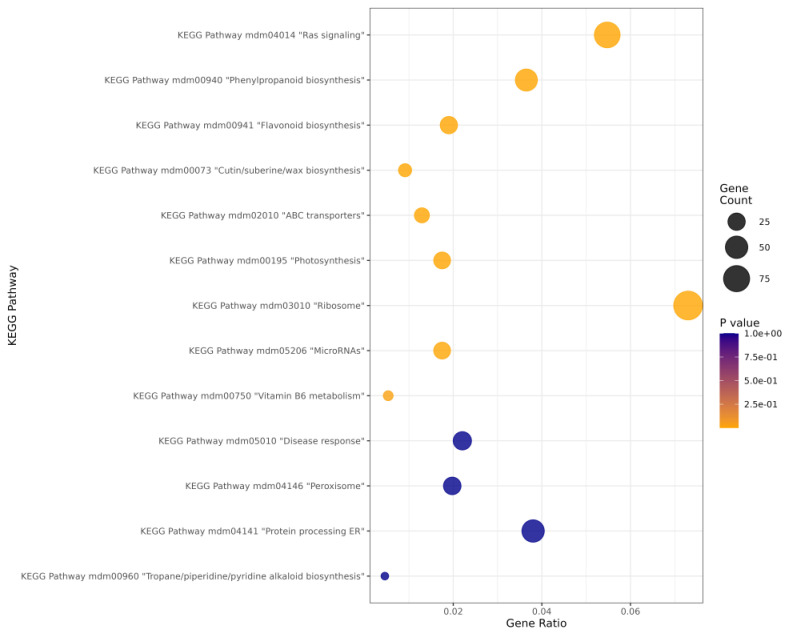
A bubble plot of the selected and most enriched KEGG pathways among DEGs; point size denotes gene count, color denotes the *p*-value and the x-axis denotes the gene ratio.

**Figure 5 ijms-26-10987-f005:**
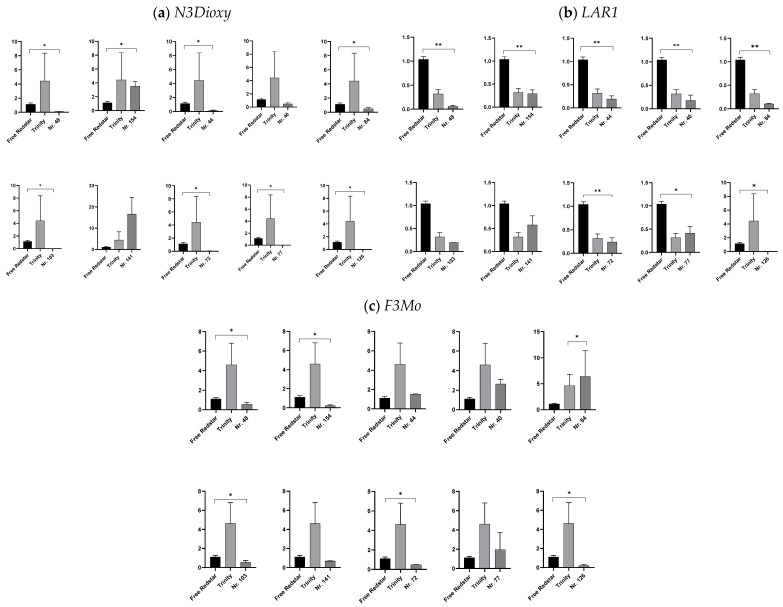
The expression profiles of genes uncovered via NGS analysis and assigned to flavonoid biosynthesis. The data show the relative fold change in the number of transcripts of genes of interest calculated as the standard error of the mean in comparison to the reference gene 18sRNA. The data were normalized in accordance with the white-fleshed cv. ‘Free Redstar’ representing ratio 1, with the significance levels of * *p* < 0.05, ** *p* < 0.01,. The relative fold change in number of gene transcripts is designated by Y axis. (**a**) *N3Dioxy*: naringenin-3-dioxygenase; (**b**) *LAR1*: leucoanthocyanidin reductase; (**c**) *F3Mo*: flavonoid 3-monooxygenase.

**Figure 6 ijms-26-10987-f006:**
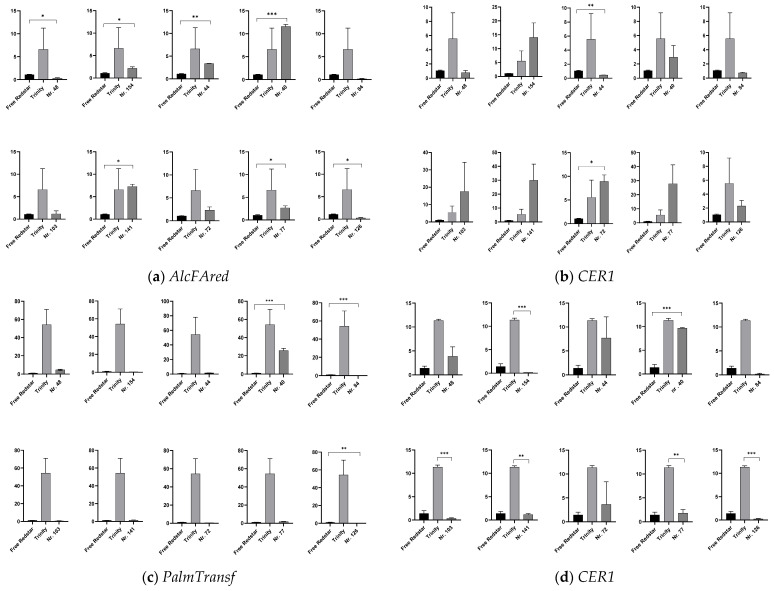
The expression profiles of genes uncovered via NGS analysis and assigned to cutin, suberine and wax synthesis. The data show the relative fold change in the number of transcripts of genes of interest calculated as the standard error of the mean in comparison to the reference gene *18sRNA*. The data were normalized in accordance with the white-fleshed cv. ‘Free Redstar’ representing ratio 1, with significance levels of * *p* < 0.05, ** *p* < 0.01 and *** *p* < 0.001. The relative fold change in number of gene transcripts is designated by Y axis. (**a**) *AlcFAred*: alcohol-forming fatty acid acyl-CoA reductase; (**b**) *CER1*: aldehyde decarbonylase; (**c**) *PalmTransf*: omega-hydroxypalmitate O-feruloyl transferase; (**d**) *CYP86A4*: fatty acid hydrolase.

**Figure 7 ijms-26-10987-f007:**
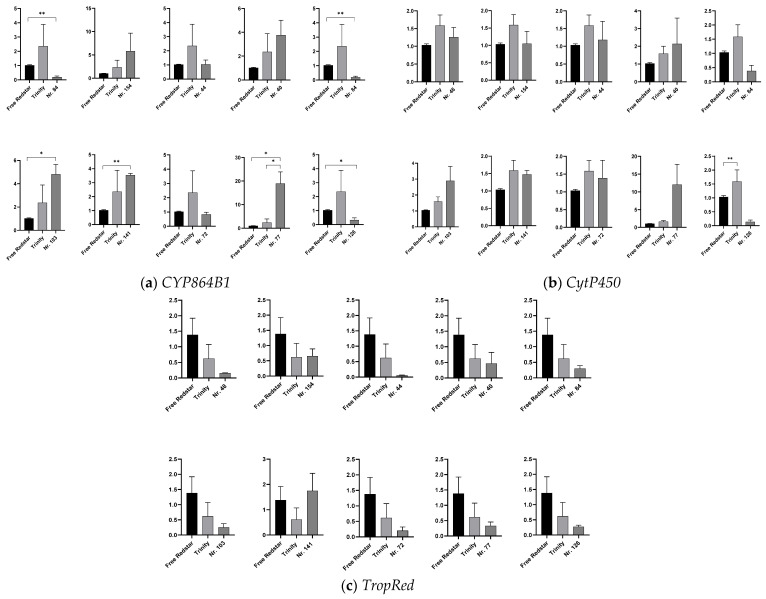
The expression profiles of genes uncovered via NGS analysis and assigned to peroxisome and tropane piperidine and pyridine alkaloid biosynthesis. The data show the relative fold changes in the number of transcripts of genes of interest calculated as the standard error of the mean in comparison to the reference gene *18sRNA*. The data were normalized in accordance with the white-fleshed cv. ‘Free Redstar’ representing ratio 1, with significance levels of * *p* < 0.05 and ** *p* < 0.01,. The relative fold change in number of gene transcripts is designated by Y axis. (**a**) *CYP865B1*: cytochrome P450 B1; (**b**) *CytP450*: cytochrome P450; (**c**) *TropRed*: tropione reductase.

**Figure 8 ijms-26-10987-f008:**
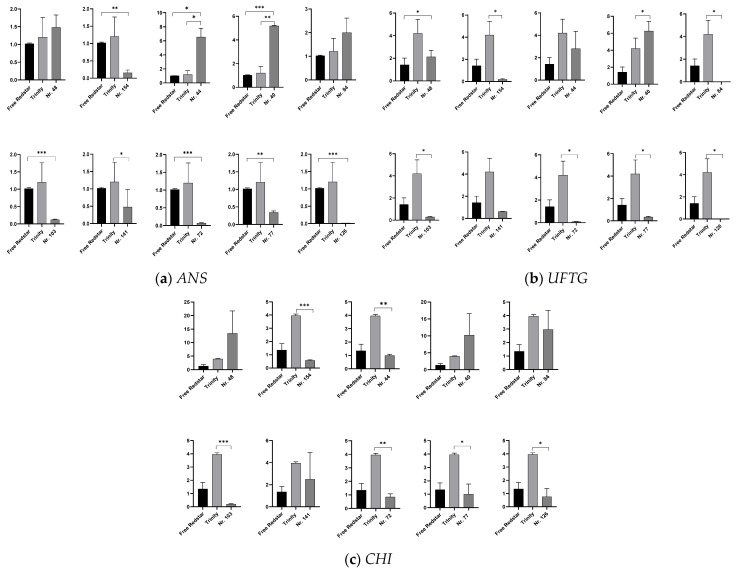
Expression profiling of structural genes involved in the biosynthesis of anthocyanins. The data show the relative fold changes in the number of transcripts of genes of interest calculated as the standard error of the mean in comparison to the reference gene *18sRNA*. The data were normalized in accordance with the white-fleshed cv. ‘Free Redstar’ representing ratio 1, with significance levels of * *p* < 0.05, ** *p* < 0.01 and *** *p* < 0.001. The relative fold change in number of gene transcripts is designated by Y axis. (**a**) *ANS*: anthocyanidin synthase; (**b**) *UFGT*: flavonoid 3′-O-glucosyl transferase; (**c**) *CHI*: chalcone isomerase.

**Figure 9 ijms-26-10987-f009:**
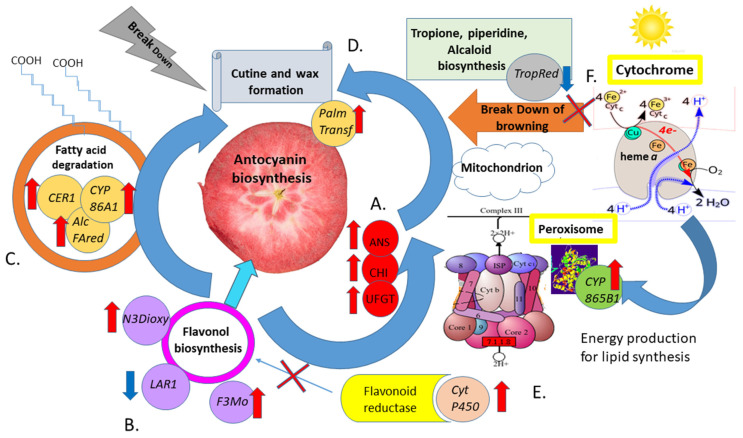
A scheme of the interaction between metabolic pathways and cell organs assisting the regulation of anthocyanin accumulation in apple fruit flesh. *N3Dioxy* (naringenin 3-dioxygenase), *LAR1* (leucoanthocyanidin reductase 1) and *F3Mo* (flavonoid 3-monooxygenase), *AlcFARed* (alcohol forming fatty acyl-CoA reductase), *CER1* (very-long-chain aldehyde decarbonylase), *PalmTransf* (omega-hydroxypalmitate O-feruloyl transferase), *CYP86A4* (fatty acid hydrolase activity), *CYT865B1*, *CytP450* (flavoprotein reductase/oxygenase activity, from cytochrome P450) and *TropRed* (tropinone reductase gene). Descriptions of potential associations between the identified pathways: (**A**) Activation of anthocyanin biosynthetic genes and determining the red color of the flesh (mdm00940). (**B**) Activation of flavonoid pathway genes (mdm00941) and the deactivation of LAR1 (flavonoid reductase). (**C**) Activation of genes regulating VLCFA synthesis (mdm00073). (**D**) Activation of omega-hydroxypalmitate O-feruloyl transferase (mdm00073) and cutin formation. (**E**) Activation of cytochrome genes with flavonoid reductase activity—originating from the proxysome (mdm04146)—associated with the production of energy required to activate the anthocyanin and lipid synthesis pathways. (**F**) Deactivation of the red-ox reaction (mdm00960)—the probable effect of light on the anthocyanin degradation process—and the inhibition of the browning reaction related to anthocyanin accumulation. Red arrows indicate up regulation, blue–down regulation of genes in identified metabolic pathways.

**Table 1 ijms-26-10987-t001:** The average anthocyanin concentrations measured for 10 apple fruits collected from each genotype. The standard error (SE) was calculated for ‘Free Redstar’.

Cultivar	Average Anthocyanin Concentration	SE Difference
Free Redstar	9.482	-
Trinity	297.472	3.078
48	321.403	6.681
154	290.723	4.779
44	184.556	5070
40	241.094	5.647
84	164.864	4643
103	77.586	2197
141	24.403	0.9361
72	20.797	1173
77	29.968	1.801
126	12.489	1.081

**Table 2 ijms-26-10987-t002:** A correlation matrix depicting the significance (*t*-test significance calculation levels of * *p* < 0.05, ** *p* < 0.01, *** *p* < 0.001, **** *p* < 0.0001) of variance calculated between analyzed genotypes on the basis of total anthocyanin accumulation in the fruit flesh of apples (GraphPad Prism 10.3). Stars in bold represent the lower significance, explaining bigger level of similarity in anthocyanin concentration between evaluated fruit samples.

	Free Redstar	Trinity	48	154	44	40	84	103	141	72	77	126
Free Redstar		****	****	****	****	****	****	****	****	****	****	*****
Trinity			*****	**ns**	****	****	****	****	****	****	****	****
48				*******	****	****	****	****	****	****	****	****
154					****	*******	****	****	****	****	****	****
44						****	******	****	****	****	****	****
40							****	****	****	****	****	****
84								****	****	****	****	****
103									****	****	****	****
141										******	*****	****
72											******	*******
77												****

**Table 3 ijms-26-10987-t003:** Oligo sequences developed in our study were used to evaluate the expression profiles of genes of interest.

Gen ID	Gene Acronym Used in This Study	Gene Function	Oligo Sequence	
			3′	5′
LOC103400025	*N3Dioxy*	naringenin 3-dioxygenase	ggcttcatcgtgtccagtca	gcctgctgctgtttgagttc
LOC103402727	*LAR1*	leucoanthocyanidin reductase	gatgtggacagggctgatcc	agccatcgaagcactcatcc
LOC103403397	*CYP865B1*	cytochrome P450 86B1, flavoprotein reductase activity	tagcagcctcttttgcgtca	atccgcaaactcgtccactt
LOC103422716	*Cyt450*	cytochrome P450 oxydase activity	tagtggaggaattggcaggg	tggctctccaggacgtctta
LOC103428452	*CER1*	very-long-chain aldehyde decarbonylase CER1-like	gacacttacctggggctacg	catctggcgattcctcctcc
LOC103437875	*F3Mo*	flavonoid 3′-monooxygenase	gttccccatcactctctggc	tcgaacctcttgtgcagctt
LOC103445140	*AlcFAred*	alcohol-forming fatty acyl-CoA reductase	agttatcatccgcccatccg	tgtacagctctaccatgcgc
LOC103418919	*CYP86A4*	cytochrome P450, fatty acid hydrolase activity	tcaagttactcaggccgctg	gagcaaccatcactcaccca
LOC103423436	*TropRed*	tropinone reductase	cctaaccctattcggccacc	ggagtacgctagaaaccgct
LOC103404168	*PalmTransf*	omega-hydroxypalmitate O-feruloyl transferase	cctcgaccaaaacattgcgg	atggaggagctgtcaatggc
Structural genes Kondo et al. [67]	*ANS*	anthocyanin synthase	caatttggcctcaaacacct	gagcttcaacaccaagtgct
	*UFGT*	UDP:flavonoid 3-O-glycosyltransferase	tccctttcactagccatgcaag	gtggaggatggagtttttacc
	*CHI*	chalcone isomerase	attatctctgctgggtca	gggaggagatggtcgaagga
Ref. [95]	*ACTIN*	actin protein	gactgtgaaactgcgaatggctca	catgaatcatcagagcaacgggca

## Data Availability

The original contributions presented in this study are included in the article/Appendix A. Further inquiries can be directed to the corresponding author.

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
