# Peer review of "Molecular Interaction of Genes Related to Anthocyanin, Lipid and Wax Biosynthesis in Apple Red-Fleshed Fruits"

_ijms, 2025, doi:10.3390/ijms262210987_

Round 1
Reviewer 1 Report
Comments and Suggestions for Authors
This manuscript investigates the genes associated with anthocyanin and wax relations by RNA-seq analysis. It may provide an insight into storage stability of red-fleshed apples. However, there are following questions that need to be revised.
- Sections such as introduction and discussion required to be reorganized and combined.
- Remove redundancy in image content of some figures.
- Some figures with minimal content should be moved to supplementary material, like 3c.
- Table 1 is not necessary; the information can be just provided in the result. Significance of Table2 can be added in Figure 1.
- Please check the reasonable of interactions of pathways from Figure 8.
- Please revise figures and tables following scientific standards for better understanding.
Author Response
29.10.2025
Keller-Przybylkowicz Sylwia
The National Institute of Horticultural Research;
Konstytucji 3-go Maja, 96-100 Skierniewice, Poland
International Journal of Molecular Science
Dear Managing Editor
Dear Lvy Ni
[Cover Letter]
We would like to thank editorial board for every remark that helped us to improve the current version of the manuscript.
We appreciate you for your precious time in proofing our paper and providing valuable comments. We have carefully considered the comments and accepted the corrections. We hope the manuscript after careful revisions meet your high standards.
All modifications in the manuscript have comments in current version accordingly to the track changes of word document and in fixed pdf.
Sincerely,
Sylwia Keller-Przybylkowicz,
Sylwia.Keller@inhort.pl
The National Institute of Horticultural Research
Konstytucji 3-go Maja 1/3
96-100 Skierniewice
Poland
Author’s responses to the comments of the reviewer #1 of the manuscript entitled “Molecular interaction of genes related with anthocyanin, lipid and wax biosynthesis in apple red-fleshed fruits., submitted to International Journal of Molecular Science under the tracking number: ijms-3954842.
- Sections such as introduction and discussion required to be reorganized and combined.
Thank You for the suggestion. The added paragraphs and some rereangment made in introduction and discussion give the better understsnding of the research problem.
The remark has been accepted. The introduction and discusion were fulfilled accroding to the reviewr suggestions.
The paragraph of introduction was:
Plant secondary metabolites such as flavonoids, anthocyanins, terpenoids, long chain fatty acids and alkaloids, have huge impact on the apple fruit development, fruit physiological parameters and quality. They also play an important role in coordination of the interactions between plants and environment, as well as are involved in plant tissue protection [1–4]. Genetic, transcriptome and environmental factors (such as sugars metabolism or plant hormones, as well as light, temperature, water), significantly affect anthocyanin biosynthesis, resulting in color development of apple fruit skin and flesh [5–7]. In accordance to anthocyanins health-promoting properties, there is an increase of interest in breeding of red-fleshed fruits in various crops like: sweet orange, peach, kiwi [8,9] etc.
The breeding of apple red-fleshed fruit trait efforts can be traced back to the XVIIth centuries, when the unusual wild apple species M. pumila var. niedzwetzkyana were discovered in Turkestan forests. Centuries of its observations underlined significant correlation of M. sieversii (M. domestica ancestor) with wild M. pumila var. niedzwetzkyana, which develops fruits with red-colored skin, flesh and blossoms [10,11].
Applying M. pumila var. niedzwetzkyana in further breeding programs yielded a series of red-fleshed apple cultivars, being designated as Type I [12,13]. In this case, the overexpression of MYB10 is strongly correlated with the regulation of red apple flesh color through upload anthocyanin accumulation. Apple cultivars in which red-fleshed phenotype do not co-segregate with MYB10 - described as type II of anthocyanin regulation, produce white-fleshed fruit [2, 10, 14, 15]. There are numerous uncovered transcription factors (TFs), involved in anthocyanin pathway regulation [16, 17]. TFs carrying structurally conserved DNA-binding domain, consisting of two-repeats of R2R3, are strongly associated with the anthocyanin biosynthesis. Further studies have also confirmed the dependence of specific MYBs on binding coregulatory proteins such as: bHLH and WD40, forming the specific domain complexes to activate transcription of anthocyanidin structural genes, acting either early – EABG (PAL, C4H, 4CL, CHS, CHI and F3H) or late - LABG (DFR, ANS and UFGT) in biosynthesis pathway [18–22].
Thanks to the changes in breeding objectives of apple characteristics from ornamental and processing features to functional fruit traits and fresh products [23–29], a series of red-fleshed apple cultivars: such as ‘jpp35’, ‘Weirouge’, ‘Baya Marisa’, ‘Redlove’, and ‘Meihong’ has been released. They can be introduced as valuable material with desirable health-promoting properties. Nevertheless, due to the characteristic flesh color and the unfavorable balance between the acid and sugar content, they are still considered as not satisfactorily attractive products for direct consumption [30].
Next factor impacting the apple fruit attractiveness is composition of the cuticle (the waxy layer of the fruit's skin), important for maintaining the fruit's long-term storage capacity [31]. Recent studies of fruit skin of "Golden Delicious" (white-skinned fruits) and "Red Delicious" (dark red-skinned fruits) cv., revealed varying levels of hydrolyzed cutin molecules and a difference in composition of aliphatic hydrocarbons. Their comparison biochemical analysis showed a significantly low content of polysaccharides and phenolic compounds in skin cuticle of 'Red Delicious' [32].
Many authors approve sugars as an important precursors in anthocyanin production, impacting cutine formation, and serve as signaling substances that can induce their cytoplasmic biosynthesis [33, 34]. They promote glycosylation of unstable anthocyanidins by UDP-Glc: flavonoid-3-O-glucosyltransferase (UFGT) to stable colorful (pink to purple) anthocyanins [35, 36, 37]. In Arabidopsis sucrose and sucrose transporters (known as SUCs) are able to regulate production of anthocyanin pigment factors (PAP) followed by expression of DFR (dihydroflavonol - 4 reductase) structural gene, and finally induce anthocyanin synthesis and promote their accumulation [38-42]. It was also observed that sucrose can activate MdHXK1 gene - encoding hexokinase, which phosphorylates the MdbHLH3 transcription factor and mediate apple fruit coloration [43].
Very long chain fatty acids (VLCFA) are pivotal in apple skin cuticle formation. In their biosynthesis an external metabolic pathways; like tricarboxylic acid cycle (TCA) and glycolysis, which regulate carbon and acetyl CoA production providing adequate precursors for fatty acid elongation are employed [44-49]. Additionally, the significant role in cuticle development, deposition and composition have cell wall invertase gene (CWI). Generally, CWI cleaves sucrose, irreversibly yielding glucose and fructose, which can be tracked via hexose transporter. These hexose sugars, are than taken up by plant cells via hexose transporters and also may act as signaling molecules [50–52].
As red flesh color of apple fruits and cutin formation depend upon a particular combination of different molecular factors, current genetic models do not explain the observed variations in apple flesh pigmentation intensity and its interlinkage among the fatty acids and wax biosynthesis.
In our study we have uncovered new genes, involved in alkaloids and general fatty acid biosynthesis, probably contributing in final fruit wax coating process. Their expression profile was validated in the fruit flesh of selected hybrid genotypes, derived from ‘Free Redstar’ and ‘Trinity’ crosspollination, differ in fruit flesh coloration (white and red respectively). This research may shed new light on the complexity of the process of anthocyanin accumulation and degradation, confirming the relatedness of anthocyanin formation and final cuticle disintegration in red-fleshed apple fruits.
Now it sounds:
Plant secondary metabolites, particularly flavonoids and anthocyanins, significantly impact apple fruit development and quality while coordinating plant-environment interactions and providing tissue protection [1-4]. Environmental factors including light, temperature, and sugar metabolism substantially affect anthocyanin biosynthesis, determining color development in both apple fruit skin and flesh [5-7].
Given anthocyanins' health-promoting properties and increasing consumer demand for functional foods, breeding programs worldwide have intensified efforts to develop red-fleshed fruit varieties across multiple crops, including sweet orange, peach, and kiwi [8,9]. Moreover, having in mind, massive apple fruit production industry (annual global production have reached over 80 ml tons), its management raises challenges in terms of cultivation and storage. An opportunity to address this situation lies in increasing the consumption of fresh apples and their products, which can be achieved by introducing into the apple industry red-fleshed varieties [7]. However, economic disadvantage is that red-fleshed varieties is relatively shorter shelf life (20-40%) compared to conventional apples, limiting distribution and increasing post-harvest losses. Understanding the molecular basis of this storage instability - particularly the relationship between anthocyanin accumulation and cuticle integrity could be crucial for developing commercially viable red-fleshed cultivars that meet both nutritional and practical market requirements [4].
In apples, red-fleshed fruit breeding objective traces back to the 17th century discovery of Malus pumila var. niedzwetzkyana in Turkestan forests, leading to development of red-fleshed cultivars with enhanced nutritional profiles [10-15]. There are numerous uncovered transcription factors (TFs), involved in anthocyanin pathway regulation [16, 17]. TFs carrying structurally conserved DNA-binding domain, consisting of two-repeats of R2R3, are strongly associated with the anthocyanin biosynthesis. Further studies have also confirmed the dependence of specific MYBs on binding coregulatory proteins such as: bHLH and WD40, forming the specific domain complexes to activate transcription of anthocyanidin structural genes, acting either early - EABG (PAL, C4H, 4CL, CHS, CHI and F3H) or late - LABG (DFR, ANS and UFGT) in biosynthesis pathway [18-22].
Thanks to the changes in breeding objectives of apple characteristics from ornamental and processing features to functional fruit traits and fresh products [23-29], a series of red-fleshed apple cultivars: such as ‘jpp35’, ‘Weirouge’, ‘Baya Marisa’, ‘Redlove’, and ‘Meihong’ has been released. They can be introduced as valuable material with desirable health-promoting properties. Despite centuries of breeding efforts, red-fleshed apple varieties remain commercially limited due to unfavorable acid-sugar balance and concerns about storage stability [30].
Next factor impacting the apple fruit attractiveness is composition of the cuticle (the waxy layer of the fruit's skin), important for maintaining the fruit's long-term storage capacity [31]. Recent studies of fruit skin of "Golden Delicious" (white-skinned fruits) and "Red Delicious" (dark red-skinned fruits) cv., revealed varying levels of hydrolyzed cutin molecules and a difference in composition of aliphatic hydrocarbons. Recent comparative studies of 'Golden Delicious' (white-skinned) and 'Red Delicious' (red-skinned) revealed differential cuticle composition, with significantly reduced polysaccharides and phenolic compounds in red varieties [32]. This suggests a potential molecular link between pigmentation and fruit storage characteristics that remains poorly understood. Many authors approve sugars as an important precursors in anthocyanin production, impacting cutine formation, and serve as signaling substances that can induce their cytoplasmic biosynthesis [33, 34]. They promote glycosylation of unstable anthocyanidins by UDP-Glc: flavonoid-3-O-glucosyltransferase (UFGT) to stable colorful (pink to purple) anthocyanins [35, 36, 37]. In Arabidopsis sucrose and sucrose transporters (known as SUCs) are able to regulate production of anthocyanin pigment factors (PAP) followed by expression of DFR (dihydroflavonol - 4 reductase) structural gene, and finally induce anthocyanin synthesis and promote their accumulation [38-42]. It was also observed that sucrose can activate MdHXK1 gene - encoding hexokinase, which phosphorylates the MdbHLH3 transcription factor and mediate apple fruit coloration [43].
Very long chain fatty acids (VLCFA) are pivotal in apple skin cuticle formation. In their biosynthesis an external metabolic pathways; like tricarboxylic acid cycle (TCA) and glycolysis, which regulate carbon and acetyl CoA production providing adequate precursors for fatty acid elongation are employed [44-49]. Additionally, the significant role in cuticle development, deposition and composition have cell wall invertase gene (CWI). Generally, CWI cleaves sucrose, irreversibly yielding glucose and fructose, which can be tracked via hexose transporter. These hexose sugars, are than taken up by plant cells via hexose transporters and also may act as signaling molecules [50-52].
Current genetic models inadequately explain observed variations in apple flesh pigmentation intensity and its relationship with fatty acid and wax biosynthesis pathways. This knowledge gap limits both breeding efficiency and our understanding of factors affecting fruit quality and shelf life.
In our study we have uncovered new genes, involved in alkaloids and general fatty acid biosynthesis, probably contributing in final fruit wax coating process. Their expression profile was validated in the fruit flesh of selected hybrid genotypes, derived from ‘Free Redstar’ and ‘Trinity’ crosspollination, differ in fruit flesh coloration (white and red respectively). This research may shed new light on the complexity of the process of anthocyanin accumulation and degradation, confirming the relatedness of anthocyanin formation and final cuticle disintegration in red-fleshed apple fruits
In the paragraph of discussion we heve made some proper changes.
It was:
3.2. Confirmation of contribution of external-pathways genes in anthocyanin biosynthesis
The relation of flavonoids and cutin formation was firstly described in tomato fruits by Heredia and coworkers [56]. They have explained that some of flavonoids are accumulated in cell vacuoles, and others are de novo synthesized and incorporated into the cuticle [56]. This indicate, that accumulation of anthocyanins resulting in lower level of cutin formation, following the looseness of apple shelf life ability.
Since the flavanones/anthocyanins and cutin relations have not been yet reported in apples, in this research we have confirmed the complex mechanism of anthocyanin accumulation, demand outside metabolites to be activated.
In this issue, four genes: AlcFARed (alcohol forming fatty acyl-CoA reductase), CER1 (aldehyde decarbonylase), PalmTransf (omega-hydroxypalmitate O-feruloyl transferase) and CYP86A1 (fatty acid hydrolase) mapped on cutin and waxy biosynthesis pathway (KEGG: mdm00073) and three: CYP865B1, CytP450 (probably flavoprotein reductase from cytochrome P450) mapped on peroxisome (KEGG: mdm04146) and TropRed (Tropione reductase) mapped on tropane piperidine and pyridine alkaloid biosynthesis (KEGG: mdm00960) were uncovered to be differentially expressed in red-fleshed fruits of ‘Trinity’.
Apple cuticle is composed of very long-chain fatty acids (VLCFAs, typically between C20–C34) derived from alkanes, alcohols, esters, aldehydes, ketones and triterpenoids [74, 75]. They are de novo synthesized in plastids under the catalysis of fatty acid (FSA) synthase complex (FAS) generally consisting fatty acyl-ACP thioesterase (FAT) and fatty acid elongase (FAE). In contrast long chain fatty acids are degradeted by acyl-reduction and decarboxylation pathways and mediated by fatty acid transferases [76], and fatty acid hydrolase (CYP86A) (Höfer et al. 2008).
Our study underlined the higher activity of AlcFARed, CER1, PalmTransf and CYP86A4 (recognized and FA reductases, transformation of VLCFA, mapped in cutin biosynthesis pathway (KEGG:mdm00073)) in red-fleshed apple fruits and allowed us to uncover the specific relation of significant reduction of VLCFs released in fruits reached in anthocyanins.
A group of molecules employed in cutin forming are triterpenoids, derived from isopentenyl pyrophosphate (IPP, C5). They play a special role in plant cutin layer formation and proceeding with acetyl-CoA (energy precursor in different tissues) [78, 52]. There are also other mechanisms applied into triterpenoids synthesis such as squalene cyclization, hydroxylation, glycosylation, and other structural modifications in which oxidosqualene cyclases (OSC), cytochrome P450 monooxygenases (CYP), and glycosyltransferases (UGTs), which seems to be pivotal in cuticle formation [75]. Our data seems to confirm this observations, thus we have observed significant activation of genes CYP and CER in the fruits of red-fleshed ‘Trinity’ (Figure 8).
Revealed in our research CER1 gene was assigned to the group of CER-like genes previously discovered in different species such as Arabidopsis [79], tomato [80], sweet cherry [81] and orange [82]. Trivedi and coworkers have underlined their skin-specific expression, suggesting that this genes might be responsible for the differential accumulation of very long chain aliphatic compounds [31]. As it was suggested CER genes play important role in alkane biosynthesis, as being linked to aldehyde biosynthesis process and VLCFAs decarbonylation pathway [81, 82]. Another type of genes in fatty acid transformation (such as omega-hydroxypalmitate O-feruloyl transferase) and reductases (like alcohol forming fatty acyl-CoA reductase) and fatty acid hydrolases, generally hampering alcohols production, necessary for fatty acid elongation [76, 52]. CER/FAR interaction, which was firstly determined for Arabidopsis, postulate, that both generate the primary alcohols and alkanes, finally associated with cuticular wax formation [83, 84]. Our results confirm this relation, and highlights this mechanism for the first time in red fleshed apple fruits.
Since the final regulations of phenylopropanoids biosynthesis are energy-intensive, there are several standalone oxidoreductases (CYP) responsible for their molecular transformation, catalyzed in cell structures such as cytochromes and peroxisomes, which up taking the secondary metabolites derived from aromatic amino acids phenylalanine in most plants [85]. As a result of these reactions, different fatty acid conjugates, plant hormones, secondary metabolites, lignin, and many protective chemicals are produced [86–88].
In the membranes of the endoplasmic reticulum, electrons are transferred directly from NADPH to cytochrome via the NADPH-cytP450 reductase complex of flavoprotein, anchored to one layer of the membrane by a hydrophobic chain. For some enzymatic molecules of cytochrome P450, such as cytochrome b5 and cytb5-cytP450 reductase, also a flavoprotein, may participate in the general electron transfer in the flavonoid cycle [89]. The genes CytP450 belonging to the flavoprotein reductases discovered in our study seems to block the flavonoid reductase thus accelerate the anthocyanin biosynthesis. This interaction in red fleshed apple fruits have been explained for the first time.
The crucial enzyme, which functionally catalyst the synthesis of the intermediate 4-(1-methyl-2-pyrrolidinyl)-3-oxobutanoic acid, finally transformed into tropinone through the catalytic activity of the cytochrome P450 is tropinone synthase (CYP82M3) [90].
The biosynthetic pathway of tropinone was fully elucidated in belladonna and it suggests its role in flower petal pigmentation [91]. In our study we have discovered the negative correlation between down regulated TropRed gene (mdm00960), and up regulated Cyp865B1 and CYP86A4 (mdm00073), leading to the activation of alkaloid biosynthesis necessary for final fatty acid elongation (Figure 8.).
Now its sounds:
3.2. Confirmation of contribution of external-pathways genes in anthocyanin biosynthesis
The relation of flavonoids and cutin formation was firstly described in tomato fruits by Heredia and coworkers [56]. They have explained that some of flavonoids are accumulated in cell vacuoles, and others are de novo synthesized and incorporated into the cuticle [56]. This indicates that accumulation of anthocyanins resulting in lower level of cutin formation, following the looseness of apple shelf life ability.
Since the flavanones/anthocyanins and cutin relations have not been yet reported in apples, in this research we have confirmed the complex mechanism of anthocyanin accumulation, demand outside metabolites to be activated.
In this issue, four genes: AlcFARed (alcohol forming fatty acyl-CoA reductase), CER1 (aldehyde decarbonylase), PalmTransf (omega-hydroxypalmitate O-feruloyl transferase) and CYP86A1 (fatty acid hydrolase) mapped on cutin and waxy biosynthesis pathway (KEGG: mdm00073) and three: CYP865B1, CytP450 (probably flavoprotein reductase from cytochrome P450) mapped on peroxisome (KEGG: mdm04146) and TropRed (Tropione reductase) mapped on tropane piperidine and pyridine alkaloid biosynthesis (KEGG: mdm00960) were uncovered to be differentially expressed in red-fleshed fruits of ‘Trinity’.
Apple cuticle is composed of very long-chain fatty acids (VLCFAs, typically between C20–-C34) derived from alkanes, alcohols, esters, aldehydes, ketones and triterpenoids [754, 765]. They are de novo synthesized in plastids under the catalysis of fatty acid (FSA) synthase complex (FAS) generally consisting fatty acyl-ACP thioesterase (FAT) and fatty acid elongase (FAE). In contrast long chain fatty acids are degradeted by acyl-reduction and decarboxylation pathways and mediated by fatty acid transferases [776], and fatty acid hydrolase (CYP86A) (Höfer et al. 2008). Our study underlined the higher activity of AlcFARed, CER1, PalmTransf and CYP86A4 (recognized and FA reductases, transformation of VLCFA, mapped in cutin biosynthesis pathway (KEGG:mdm00073)) in red-fleshed apple fruits and allowed us to uncover the specific relationship between a significant reduction in the amount of VLCF released in fruits and anthocyanin contentspecific relation of significant reduction of VLCFs released in fruits reached in anthocyanins.
A group of molecules employed in cutin forming are triterpenoids, derived from isopentenyl pyrophosphate (IPP, C5). They play a special role in plant cutin layer formation and proceeding with acetyl-CoA (energy precursor in different tissues) [788, 52]. There are also other mechanisms applied into triterpenoids synthesis such as squalene cyclization, hydroxylation, glycosylation, and other structural modifications in which oxidosqualene cyclases (OSC), cytochrome P450 monooxygenases (CYP), and glycosyltransferases (UGTs), which seems to be pivotal in cuticle formation [765]. Our data seems to confirm this observations, thus we have observed noted significant activation of genes CYP and CER in the fruits of red-fleshed ‘Trinity’ (Figure 98).
Revealed in our research CER1 gene was assigned to the group of CER-like genes previously discovered in different species such as Arabidopsis [79], tomato [80], sweet cherry [81] and orange [82]. Trivedi and coworkers have underlined their skin-specific expression, suggesting that this genes might be responsible for the differential accumulation of very long chain aliphatic compounds [31]. As it was suggested CER genes play important role in alkane biosynthesis, as being linked to aldehyde biosynthesis process and VLCFAs decarbonylation pathway [81, 82]. Another type of genes in fatty acid transformation (such as omega-hydroxypalmitate O-feruloyl transferase) and reductases (like alcohol forming fatty acyl-CoA reductase) and fatty acid hydrolases, generally hampering alcohols production, necessary for fatty acid elongation [76, 52]. CER/FAR interaction, which was firstly determined for Arabidopsis, postulate, that both generate the primary alcohols and alkanes, finally associated with cuticular wax formation [83, 84]. Our results confirm this relation, and highlights this mechanism for the first time in red -fleshed apple fruits.
One of the main factor impacting the regulatory mechanism in plant cytochrome and peroxisome is light. This concludes the very intensive connections of those important cell organelles. Since light is essential for plant growth and development some records underline their affection in anthocyanin biosynthesis, (negatively in apple fruits). Generally activated photons inhibits the transcription factors ability [85]. This mechanism have not been investigated yet, and we have retrieved some probable connections between anthocyjanins and wax biosynthesis as well as energy accumulation.
Since the final regulations of phenylopropanoids biosynthesis are energy-intensive, there are several standalone oxidoreductases (CYP) responsible for their molecular transformation, catalyzed in cell structures such as cytochromes and peroxisomes, which up taking the secondary metabolites derived from aromatic amino acids phenylalanine in most plants [895].
As a result of these reactions, different fatty acid conjugates, plant hormones, secondary metabolites, lignin, and many protective chemicals are produced [86–-88].
In the membranes of the endoplasmic reticulum, electrons are transferred directly from NADPH to cytochrome via the NADPH-cytP450 reductase complex of flavoprotein, anchored to one layer of the membrane by a hydrophobic chain.
For some enzymatic molecules of cytochrome P450, such as cytochrome b5 and cytb5-cytP450 reductase, also a flavoprotein, may participate in the general electron transfer in the flavonoid cycle [8990]. The genes CytP450 belonging to the flavoprotein reductases discovered in our study seems to block the flavonoid reductase thus accelerate the anthocyanin biosynthesis. This interaction in red -fleshed apple fruits have been explained for the first time.
The crucial enzyme, which functionally catalyst the synthesis of the intermediate 4-(1-methyl-2-pyrrolidinyl)-3-oxobutanoic acid, finally transformed into tropinone through the catalytic activity of the cytochrome P450 is tropinone synthase (CYP82M3) [901].
The biosynthetic pathway of tropinone was fully elucidated in belladonna and it suggests its role in flower petal pigmentation [912]. In our study we have discovered the negative correlation between down regulated TropRed gene (mdm00960), and up regulated Cyp865B1 and CYP86A4 (mdm00073), leading to the activation of alkaloid biosynthesis necessary for final fatty acid elongation (Figure 98.). In case of red-flashed apple, ,tropinone reductase not fully investigated enzyme because they do not produce tropane alkaloids, thus we have observed negligible its activity in evaluated apple fruit flesh samples (Figure 9.). They seems to be related with apple browning, caused by polyphenol oxidase (PPO) enzymes [91 ]. This is also mechanism that could not be observed in red flesh of apples, that underline the red-fleshed fruits do no browning. However this mechanism must be deeply investigated.
The paragraph of Conclusions was also revised:
It was:
- Conclusions
Our analysis allowed us to discover new insights into the regulation of apple flesh color acquisition. We found an undoubted link between the anthocyanidin biosynthetic pathways and parallel metabolism involved fruit wax coating, controlled by factors, such as: N3Dioxy (naringenin 3-dioxygenase), LAR1 (leucoanthocyanidin reductase 1) and F3Mo (flavonoid 3-monooxygenase) mapped to the flavonoid synthesis pathway (KEGG: mdm00941); AlcFARed (alcohol-forming fatty acyl-CoA reductase), CER1 (aldehyde decarbonylase), PalmTransf (omega-hydroxypalmitate O-feruloyl transferase) and CYP86A1 (fatty acid hydrolase) - mapped to the cutin and wax biosynthetic pathway (KEGG: mdm00073); and CYP865B1, CytP450 (probably a cytochrome P450 flavoprotein reductase) mapped to peroxisomes (KEGG: mdm04146); and TropRed (tropione reductase) mapped to tropane piperidine and pyridine alkaloid biosynthesis (KEGG: mdm00960).
All discovered genes showed significant correlations with total anthocyanin content and visible fruit flesh color in the fruit flesh of analyzed genotypes. Although, many reports exist on anthocyanin development, both in the peel and fruit flesh, the complexity of this phenomenon is not fully understood. We confirmed that the interaction of the flavonoid biosynthetic pathway and the fatty acid degradation mechanism may overlap during fruit formation. However, the final postulates has to be validated, especially in the context of introduction of selected sequences as potential candidate genes for conducting MAS procedure in future breeding programs of red-fleshed apple trees.
Now it summarise and outlining the practical and commercial and applicable significance, as well as the role of red-fleshed apples in breeding programs.
In the final version, the conclusions paragraph sound like:
Our analysis revealed novel molecular interactions between anthocyanin biosynthesis and wax metabolism pathways in apple fruit flesh. The identification of ten key genes—N3Dioxy, LAR1, F3Mo (flavonoid pathway); AlcFARed, CER1, PalmTransf, CYP86A4 (wax biosynthesis); and TropRed, CyP865B1, CytP450 (alkaloid biosynthesis/peroxisome)—provides new insights into the molecular basis of flesh coloration and storage stability relationships.
The identified genes offer immediate potential as molecular markers for marker-assisted selection (MAS) in red-fleshed apple breeding programs. Priority should be given to developing functional markers for: CER1 and AlcFARed expression levels as predictors of storage stability, F3Mo/LAR1 ratio as an indicator of flesh color intensity. Combined marker assay for simultaneous selection of color and storage traits.
Our findings suggest that red-fleshed varieties' storage limitations stem from reduced wax biosynthesis capacity. Future research should focus on developing cultivation practices to enhance CER1 and CYP86A4 expression, investigating postharvest treatments targeting wax metabolism pathways, and finally exploring genetic modification approaches to restore wax biosynthesis in high-anthocyanin backgrounds
Successfully addressing storage limitations could unlock significant market potential for red-fleshed apples, within functional food sectors. Improved varieties would support sustainable agriculture by providing growers with premium products while delivering enhanced nutritional benefits to consumers.
Future investigations should validate these gene-trait relationships across diverse genetic backgrounds, develop high-throughput screening methods for breeding programs, and explore environmental factors influencing the anthocyanin-wax metabolism balance. Integration with genomics-assisted breeding platforms will be essential for translating these molecular insights into commercial cultivar development.
According to arrangements made in the manuscript, two reference position were added:
- Liu, H.; Liu, Z.; Wu, Y.; Zheng, L.; Zhang, G. 2021. Regulatory Mechanisms of Anthocyanin Biosynthesis in Apple and Pear. Int. J. Mol. Sci. 22, 8441. https://doi.org/10.3390/ijms22168441
And
- Birgit Dräger B. 2006. Tropinone reductases, enzymes at the branch point of tropane alkaloid metabolism, Phytochemistry, 67, 327-337; https://doi.org/10.1016/j.phytochem.2005.12.001
The references were also numerized properly, also in the cited text.
- Remove redundancy in image content of some figures.
Thank You for the suggestion. Indeed the repetition of diagram titles ware confusing.
The remark has been accepted:
The image content of figures representing expression profiles were apropriately changed.
- Some figures with minimal content should be moved to supplementary material, like 3c.
Many thanks for the sugeestion, this can be cosiderred as additional data.
The remark has been accepted:
The figure 3c was moved to the suplementarny materials.
So the numeric order of figures from suplementarny material was subsequently changed. This has been changed also in the text.
- Table 1 is not necessary; the information can be just provided in the result. Significance of Table2 can be added in Figure 1.
Thank You for the suggestion, this will avoid any calculation dublications.
The remark has been accepted:
The table numer 2 was removed and the figure 1 was properly changed.
The paragraph was:
Figure 1. Total Anthocyanin Concentration in apple fruit flesh of parental and hybrid genotypes, evaluated spectrophotometrically and compared with regard to cyanidin-3-glucoside. Data are presented as an average anthocyanin concentration with standard error of the fruit mean (±SEM) compared to ‘Free Redstar’ white fleshed fruit cv. and t-test significance calculation level p<0,05*, 0,01 **, 0,001***; GraphPad Prism 10.3
Table 1. Summary of the statistical analysis of the flesh fruit samples anthocyanin content variability.
|
ANOVA summary |
|
|
F |
1205 |
|
P-value |
<0,0001 |
|
P-value summary |
**** |
|
Significant diff. among means (P < 0.05) |
Yes |
|
R 2 |
0,9920 |
Table 2. Correlation matrix in accordance to significance of variance between analyzed genotypes on the basis in total anthocyanin accumulation in fruit flesh apples.
|
Free Redstar |
Trinity |
48 |
154 |
44 |
40 |
84 |
103 |
141 |
72 |
77 |
126 |
|
|
Free Redstar |
**** |
**** |
**** |
**** |
**** |
**** |
**** |
**** |
**** |
**** |
* |
|
|
Trinity |
* |
ns |
**** |
**** |
**** |
**** |
**** |
**** |
**** |
**** |
||
|
48 |
*** |
**** |
**** |
**** |
**** |
**** |
**** |
**** |
**** |
|||
|
154 |
**** |
*** |
**** |
**** |
**** |
**** |
**** |
**** |
||||
|
44 |
**** |
** |
**** |
**** |
**** |
**** |
**** |
|||||
|
40 |
**** |
**** |
**** |
**** |
**** |
**** |
||||||
|
84 |
**** |
**** |
**** |
**** |
**** |
|||||||
|
103 |
**** |
**** |
**** |
**** |
||||||||
|
141 |
** |
* |
**** |
|||||||||
|
72 |
** |
*** |
||||||||||
|
77 |
**** |
Now the part of this paragraph looks like:
Figure 1. Total Anthocyanin Concentration in apple fruit flesh of parental and hybrid genotypes, evaluated spectrophotometrically and compared with regard to cyanidin-3-glucoside. Data are presented as an average anthocyanin concentration with standard error of the fruit mean (±SEM) compared to ‘Free Redstar’ white-fleshed fruit cv. and t-test significance calculation levels of P < 0.05*, P < 0.01**, P < 0.001***; P < 0,0001****, GraphPad Prism 10.3.
Table 1. Correlation matrix in accordance to significance of variance between analyzed genotypes on the basis in total anthocyanin accumulation in fruit flesh apple, t -test significance calculation levels of P < 0.05*, P < 0.01**, P < 0.001***; P < 0,0001****, GraphPad Prism 10.3.
|
|
Free Redstar |
Trinity |
48 |
154 |
44 |
40 |
84 |
103 |
141 |
72 |
77 |
126 |
|
Free Redstar |
|
**** |
**** |
**** |
**** |
**** |
**** |
**** |
**** |
**** |
**** |
* |
|
Trinity |
|
|
* |
ns |
**** |
**** |
**** |
**** |
**** |
**** |
**** |
**** |
|
48 |
|
|
|
*** |
**** |
**** |
**** |
**** |
**** |
**** |
**** |
**** |
|
154 |
|
|
|
|
**** |
*** |
**** |
**** |
**** |
**** |
**** |
**** |
|
44 |
|
|
|
|
|
**** |
** |
**** |
**** |
**** |
**** |
**** |
|
40 |
|
|
|
|
|
|
**** |
**** |
**** |
**** |
**** |
**** |
|
84 |
|
|
|
|
|
|
|
**** |
**** |
**** |
**** |
**** |
|
103 |
|
|
|
|
|
|
|
|
**** |
**** |
**** |
**** |
|
141 |
|
|
|
|
|
|
|
|
|
** |
* |
**** |
|
72 |
|
|
|
|
|
|
|
|
|
|
** |
*** |
|
77 |
|
|
|
|
|
|
|
|
|
|
|
**** |
Authors decided to keep the table with correlation matric, du to it shows the significant relatnes between each fruit samples, but some differential of significance was added to the diagram.
- Please check the reasonable of interactions of pathways from Figure 8.
The ramark has been accepted:
The figure 8 was previously incorrectly numerized by mistake, it suppose to be figure 9.
The remark has been accepted.
The figure was changed in accordance to present the reliable interaction between identified pathways.
Additional description of the figure was also added.
The scheme was previously presented like:
Figure 8. Scheme of interaction between metabolic pathways and cell organs assisting the regulation of anthocyanin accumulation in apple fruit flesh. N3Dioxy (naringenin 3-dioxygenase), LAR1 (leucoanthocyanidin reductase 1) and F3Mo (flavonoid 3-monooxygenase), AlcFARed (alcohol forming fatty acyl-CoA reductase), CER1 (very-long-chain aldehyde decarbonylase), PalmTransf (omega-hydroxypalmitate O-feruloyl transferase), CYP86A4 (fatty acid hydrolase activity), CYT865B1, CytP450 (flavoprotein reductase/oxygenase activity, from cytochrome P450), TropRed (tropinone reductase gene)
Now it looks like:
Figure 9. Scheme of interaction between metabolic pathways and cell organs assisting the regulation of anthocyanin accumulation in apple fruit flesh. N3Dioxy (naringenin 3-dioxygenase), LAR1 (leucoanthocyanidin reductase 1) and F3Mo (flavonoid 3-monooxygenase), AlcFARed (alcohol forming fatty acyl-CoA reductase), CER1 (very-long-chain aldehyde decarbonylase), PalmTransf (omega-hydroxypalmitate O-feruloyl transferase), CYP86A4 (fatty acid hydrolase activity), CYT865B1, CytP450 (flavoprotein reductase/oxygenase activity, from cytochrome P450), TropRed (tropinone reductase gene);
Description of potential associations of identified pathways.
- Activation of anthocyanin biosynthetic genes - determining the red color of the flesh (mdm00940). A.
- Activation of flavonoid pathway genes (mdm00941) - deactivation of LAR1 (flavonoid reductase).
- Activation of genes regulating VLCFA synthesis (mdm00073).
- Activation of omega-hydroxypalmitate O-feruloyl transferase (mdm00073) - cutin formation.
- Activation of cytochrome genes with flavonoid reductase activity - originating from the proxysome (mdm04146) - associated with the production of energy required to activate the anthocyanin and lipid synthesis pathways.
- Deactivation of the red-ox reaction (mdm00960) - the probable effect of light on the anthocyanin degradation process - inhibition of the browning reaction related with anthocyanin accumulation.
- Please revise figures and tables following scientific standards for better understanding.
Thank You for the suggestion.
The remark has been accepted.
The figures were revise in accordance to the scientific standards.

Reviewer 2 Report
Comments and Suggestions for Authors
The article “Molecular interaction of genes related with anthocyanin, lipid and wax biosynthesis in apple red-fleshed fruits” presents a well-structured technical and molecular study focused on the genetic relationship between anthocyanin accumulation and lipid or wax metabolism in apples. The topic is relevant, and the authors used appropriate molecular and bioinformatic methods, combining RNAseq data with qPCR validation. The presentation of results is clear, and the discussion provides a logical interpretation of mostly molecular aspects of the article supported by literature references.
The study brings useful observations on the possible interaction between pigment formation and wax biosynthesis pathways, which may have implications for breeding and postharvest quality. The figures and tables are informative and generally easy to follow.
While the article provides a solid description of molecular mechanisms related to anthocyanin and wax biosynthesis, it would benefit from a clearer explanation of the broader motivation behind the study. The introduction focuses mainly on genetic and biochemical aspects, but it does not sufficiently highlight why the red coloration in apple flesh is an important research topic.
A short section outlining the practical or commercial significance of red-fleshed apples—for example, their potential health benefits, consumer appeal, or role in breeding programs—would help the reader better understand the background and relevance of the research question.
Additionally, the paper would be stronger with a more explicitly formulated conclusion. The current ending summarizes the results but does not clearly state what the authors see as the next steps or main implications of their findings. The authors must improve this section by explaining how their discoveries might contribute to future breeding strategies, storage quality improvement, or the development of functional fruit varieties.
Defining a clear direction for future breeding research and underlining the societal or economic importance of this work would give the article a more complete and purposeful closing.
There are, moreover, a few minor issues that could be improved. Some parts of the text would benefit from clearer and more concise English phrasing. For example, in the abstract, the sentence “Our results, postulate, that the fatty acid degradation process is initiating in flesh of apple fruits…” should be corrected to “Our results postulate that the fatty acid degradation process is initiated in the flesh of apple fruits.”
In addition, the cultivar name “Free Redstar” is occasionally misspelled as “Free Redsar,” which should be unified.
The use of “red-fleshed” versus “red fleshed” is also inconsistent and should be standardized throughout the manuscript.
Overall, this is a well-prepared piece of research, but it needs to be improved and oriented further than only focusing on technical/laboratory part.
With minor language corrections, above metioned improvements and attention to small typographical details, the article will be ready for publication and will make a valuable contribution to the understanding of molecular mechanisms in red-fleshed apples.
Comments on the Quality of English LanguageI am not native English speaker.
Author Response
29.10.2025
Keller-Przybylkowicz Sylwia
The National Institute of Horticultural Research;
Konstytucji 3-go Maja, 96-100 Skierniewice, Poland
International Journal of Molecular Science
Dear Managing Editor
Dear Lvy Ni
[Cover Letter]
We would like to thank editorial board for every remark that helped us to improve the current version of the manuscript.
We appreciate you for your precious time in proofing our paper and providing valuable comments. We have carefully considered the comments and accepted the corrections. We hope the manuscript after careful revisions meet your high standards.
All modifications in the manuscript have comments in current version accordingly to the track changes of word document and in fixed pdf.
Sincerely,
Sylwia Keller-Przybylkowicz,
Sylwia.Keller@inhort.pl
The National Institute of Horticultural Research
Konstytucji 3-go Maja 1/3
96-100 Skierniewice
Poland
Author’s responses to the comments of the reviewer #2 of the manuscript entitled “Molecular interaction of genes related with anthocyanin, lipid and wax biosynthesis in apple red-fleshed fruits., submitted to International Journal of Molecular Science under the tracking number: ijms-3954842.
- The introduction focuses mainly on genetic and biochemical aspects, but it does not sufficiently highlight why the red coloration in apple flesh is an important research topic. A short section outlining the practical or commercial significance of red-fleshed apples—for example, their potential health benefits, consumer appeal, or role in breeding programs—would help the reader better understand the background and relevance of the research question.
Thank You for the suggestion. The added paragraphs and some rereangment made in introduction and discussion give the better understsnding of the research problem.
The remark has been accepted. The introduction and discusion were fulfilled accroding to the reviewr suggestions.
The paragraph of introduction was:
Plant secondary metabolites such as flavonoids, anthocyanins, terpenoids, long chain fatty acids and alkaloids, have huge impact on the apple fruit development, fruit physiological parameters and quality. They also play an important role in coordination of the interactions between plants and environment, as well as are involved in plant tissue protection [1–4]. Genetic, transcriptome and environmental factors (such as sugars metabolism or plant hormones, as well as light, temperature, water), significantly affect anthocyanin biosynthesis, resulting in color development of apple fruit skin and flesh [5–7]. In accordance to anthocyanins health-promoting properties, there is an increase of interest in breeding of red-fleshed fruits in various crops like: sweet orange, peach, kiwi [8,9] etc.
The breeding of apple red-fleshed fruit trait efforts can be traced back to the XVIIth centuries, when the unusual wild apple species M. pumila var. niedzwetzkyana were discovered in Turkestan forests. Centuries of its observations underlined significant correlation of M. sieversii (M. domestica ancestor) with wild M. pumila var. niedzwetzkyana, which develops fruits with red-colored skin, flesh and blossoms [10,11].
Applying M. pumila var. niedzwetzkyana in further breeding programs yielded a series of red-fleshed apple cultivars, being designated as Type I [12,13]. In this case, the overexpression of MYB10 is strongly correlated with the regulation of red apple flesh color through upload anthocyanin accumulation. Apple cultivars in which red-fleshed phenotype do not co-segregate with MYB10 - described as type II of anthocyanin regulation, produce white-fleshed fruit [2, 10, 14, 15]. There are numerous uncovered transcription factors (TFs), involved in anthocyanin pathway regulation [16, 17]. TFs carrying structurally conserved DNA-binding domain, consisting of two-repeats of R2R3, are strongly associated with the anthocyanin biosynthesis. Further studies have also confirmed the dependence of specific MYBs on binding coregulatory proteins such as: bHLH and WD40, forming the specific domain complexes to activate transcription of anthocyanidin structural genes, acting either early – EABG (PAL, C4H, 4CL, CHS, CHI and F3H) or late - LABG (DFR, ANS and UFGT) in biosynthesis pathway [18–22].
Thanks to the changes in breeding objectives of apple characteristics from ornamental and processing features to functional fruit traits and fresh products [23–29], a series of red-fleshed apple cultivars: such as ‘jpp35’, ‘Weirouge’, ‘Baya Marisa’, ‘Redlove’, and ‘Meihong’ has been released. They can be introduced as valuable material with desirable health-promoting properties. Nevertheless, due to the characteristic flesh color and the unfavorable balance between the acid and sugar content, they are still considered as not satisfactorily attractive products for direct consumption [30].
Next factor impacting the apple fruit attractiveness is composition of the cuticle (the waxy layer of the fruit's skin), important for maintaining the fruit's long-term storage capacity [31]. Recent studies of fruit skin of "Golden Delicious" (white-skinned fruits) and "Red Delicious" (dark red-skinned fruits) cv., revealed varying levels of hydrolyzed cutin molecules and a difference in composition of aliphatic hydrocarbons. Their comparison biochemical analysis showed a significantly low content of polysaccharides and phenolic compounds in skin cuticle of 'Red Delicious' [32].
Many authors approve sugars as an important precursors in anthocyanin production, impacting cutine formation, and serve as signaling substances that can induce their cytoplasmic biosynthesis [33, 34]. They promote glycosylation of unstable anthocyanidins by UDP-Glc: flavonoid-3-O-glucosyltransferase (UFGT) to stable colorful (pink to purple) anthocyanins [35, 36, 37]. In Arabidopsis sucrose and sucrose transporters (known as SUCs) are able to regulate production of anthocyanin pigment factors (PAP) followed by expression of DFR (dihydroflavonol - 4 reductase) structural gene, and finally induce anthocyanin synthesis and promote their accumulation [38-42]. It was also observed that sucrose can activate MdHXK1 gene - encoding hexokinase, which phosphorylates the MdbHLH3 transcription factor and mediate apple fruit coloration [43].
Very long chain fatty acids (VLCFA) are pivotal in apple skin cuticle formation. In their biosynthesis an external metabolic pathways; like tricarboxylic acid cycle (TCA) and glycolysis, which regulate carbon and acetyl CoA production providing adequate precursors for fatty acid elongation are employed [44-49]. Additionally, the significant role in cuticle development, deposition and composition have cell wall invertase gene (CWI). Generally, CWI cleaves sucrose, irreversibly yielding glucose and fructose, which can be tracked via hexose transporter. These hexose sugars, are than taken up by plant cells via hexose transporters and also may act as signaling molecules [50–52].
As red flesh color of apple fruits and cutin formation depend upon a particular combination of different molecular factors, current genetic models do not explain the observed variations in apple flesh pigmentation intensity and its interlinkage among the fatty acids and wax biosynthesis.
In our study we have uncovered new genes, involved in alkaloids and general fatty acid biosynthesis, probably contributing in final fruit wax coating process. Their expression profile was validated in the fruit flesh of selected hybrid genotypes, derived from ‘Free Redstar’ and ‘Trinity’ crosspollination, differ in fruit flesh coloration (white and red respectively). This research may shed new light on the complexity of the process of anthocyanin accumulation and degradation, confirming the relatedness of anthocyanin formation and final cuticle disintegration in red-fleshed apple fruits.
Now it sounds:
Plant secondary metabolites, particularly flavonoids and anthocyanins, significantly impact apple fruit development and quality while coordinating plant-environment interactions and providing tissue protection [1-4]. Environmental factors including light, temperature, and sugar metabolism substantially affect anthocyanin biosynthesis, determining color development in both apple fruit skin and flesh [5-7].
Given anthocyanins' health-promoting properties and increasing consumer demand for functional foods, breeding programs worldwide have intensified efforts to develop red-fleshed fruit varieties across multiple crops, including sweet orange, peach, and kiwi [8,9]. Moreover, having in mind, massive apple fruit production industry (annual global production have reached over 80 ml tons), its management raises challenges in terms of cultivation and storage. An opportunity to address this situation lies in increasing the consumption of fresh apples and their products, which can be achieved by introducing into the apple industry red-fleshed varieties [7]. However, economic disadvantage is that red-fleshed varieties is relatively shorter shelf life (20-40%) compared to conventional apples, limiting distribution and increasing post-harvest losses. Understanding the molecular basis of this storage instability - particularly the relationship between anthocyanin accumulation and cuticle integrity could be crucial for developing commercially viable red-fleshed cultivars that meet both nutritional and practical market requirements [4].
In apples, red-fleshed fruit breeding objective traces back to the 17th century discovery of Malus pumila var. niedzwetzkyana in Turkestan forests, leading to development of red-fleshed cultivars with enhanced nutritional profiles [10-15]. There are numerous uncovered transcription factors (TFs), involved in anthocyanin pathway regulation [16, 17]. TFs carrying structurally conserved DNA-binding domain, consisting of two-repeats of R2R3, are strongly associated with the anthocyanin biosynthesis. Further studies have also confirmed the dependence of specific MYBs on binding coregulatory proteins such as: bHLH and WD40, forming the specific domain complexes to activate transcription of anthocyanidin structural genes, acting either early - EABG (PAL, C4H, 4CL, CHS, CHI and F3H) or late - LABG (DFR, ANS and UFGT) in biosynthesis pathway [18-22].
Thanks to the changes in breeding objectives of apple characteristics from ornamental and processing features to functional fruit traits and fresh products [23-29], a series of red-fleshed apple cultivars: such as ‘jpp35’, ‘Weirouge’, ‘Baya Marisa’, ‘Redlove’, and ‘Meihong’ has been released. They can be introduced as valuable material with desirable health-promoting properties. Despite centuries of breeding efforts, red-fleshed apple varieties remain commercially limited due to unfavorable acid-sugar balance and concerns about storage stability [30].
Next factor impacting the apple fruit attractiveness is composition of the cuticle (the waxy layer of the fruit's skin), important for maintaining the fruit's long-term storage capacity [31]. Recent studies of fruit skin of "Golden Delicious" (white-skinned fruits) and "Red Delicious" (dark red-skinned fruits) cv., revealed varying levels of hydrolyzed cutin molecules and a difference in composition of aliphatic hydrocarbons. Recent comparative studies of 'Golden Delicious' (white-skinned) and 'Red Delicious' (red-skinned) revealed differential cuticle composition, with significantly reduced polysaccharides and phenolic compounds in red varieties [32]. This suggests a potential molecular link between pigmentation and fruit storage characteristics that remains poorly understood. Many authors approve sugars as an important precursors in anthocyanin production, impacting cutine formation, and serve as signaling substances that can induce their cytoplasmic biosynthesis [33, 34]. They promote glycosylation of unstable anthocyanidins by UDP-Glc: flavonoid-3-O-glucosyltransferase (UFGT) to stable colorful (pink to purple) anthocyanins [35, 36, 37]. In Arabidopsis sucrose and sucrose transporters (known as SUCs) are able to regulate production of anthocyanin pigment factors (PAP) followed by expression of DFR (dihydroflavonol - 4 reductase) structural gene, and finally induce anthocyanin synthesis and promote their accumulation [38-42]. It was also observed that sucrose can activate MdHXK1 gene - encoding hexokinase, which phosphorylates the MdbHLH3 transcription factor and mediate apple fruit coloration [43].
Very long chain fatty acids (VLCFA) are pivotal in apple skin cuticle formation. In their biosynthesis an external metabolic pathways; like tricarboxylic acid cycle (TCA) and glycolysis, which regulate carbon and acetyl CoA production providing adequate precursors for fatty acid elongation are employed [44-49]. Additionally, the significant role in cuticle development, deposition and composition have cell wall invertase gene (CWI). Generally, CWI cleaves sucrose, irreversibly yielding glucose and fructose, which can be tracked via hexose transporter. These hexose sugars, are than taken up by plant cells via hexose transporters and also may act as signaling molecules [50-52].
Current genetic models inadequately explain observed variations in apple flesh pigmentation intensity and its relationship with fatty acid and wax biosynthesis pathways. This knowledge gap limits both breeding efficiency and our understanding of factors affecting fruit quality and shelf life.
In our study we have uncovered new genes, involved in alkaloids and general fatty acid biosynthesis, probably contributing in final fruit wax coating process. Their expression profile was validated in the fruit flesh of selected hybrid genotypes, derived from ‘Free Redstar’ and ‘Trinity’ crosspollination, differ in fruit flesh coloration (white and red respectively). This research may shed new light on the complexity of the process of anthocyanin accumulation and degradation, confirming the relatedness of anthocyanin formation and final cuticle disintegration in red-fleshed apple fruits
In the paragraph of discussion we heve made some proper changes.
It was:
3.2. Confirmation of contribution of external-pathways genes in anthocyanin biosynthesis
The relation of flavonoids and cutin formation was firstly described in tomato fruits by Heredia and coworkers [56]. They have explained that some of flavonoids are accumulated in cell vacuoles, and others are de novo synthesized and incorporated into the cuticle [56]. This indicate, that accumulation of anthocyanins resulting in lower level of cutin formation, following the looseness of apple shelf life ability.
Since the flavanones/anthocyanins and cutin relations have not been yet reported in apples, in this research we have confirmed the complex mechanism of anthocyanin accumulation, demand outside metabolites to be activated.
In this issue, four genes: AlcFARed (alcohol forming fatty acyl-CoA reductase), CER1 (aldehyde decarbonylase), PalmTransf (omega-hydroxypalmitate O-feruloyl transferase) and CYP86A1 (fatty acid hydrolase) mapped on cutin and waxy biosynthesis pathway (KEGG: mdm00073) and three: CYP865B1, CytP450 (probably flavoprotein reductase from cytochrome P450) mapped on peroxisome (KEGG: mdm04146) and TropRed (Tropione reductase) mapped on tropane piperidine and pyridine alkaloid biosynthesis (KEGG: mdm00960) were uncovered to be differentially expressed in red-fleshed fruits of ‘Trinity’.
Apple cuticle is composed of very long-chain fatty acids (VLCFAs, typically between C20–C34) derived from alkanes, alcohols, esters, aldehydes, ketones and triterpenoids [74, 75]. They are de novo synthesized in plastids under the catalysis of fatty acid (FSA) synthase complex (FAS) generally consisting fatty acyl-ACP thioesterase (FAT) and fatty acid elongase (FAE). In contrast long chain fatty acids are degradeted by acyl-reduction and decarboxylation pathways and mediated by fatty acid transferases [76], and fatty acid hydrolase (CYP86A) (Höfer et al. 2008).
Our study underlined the higher activity of AlcFARed, CER1, PalmTransf and CYP86A4 (recognized and FA reductases, transformation of VLCFA, mapped in cutin biosynthesis pathway (KEGG:mdm00073)) in red-fleshed apple fruits and allowed us to uncover the specific relation of significant reduction of VLCFs released in fruits reached in anthocyanins.
A group of molecules employed in cutin forming are triterpenoids, derived from isopentenyl pyrophosphate (IPP, C5). They play a special role in plant cutin layer formation and proceeding with acetyl-CoA (energy precursor in different tissues) [78, 52]. There are also other mechanisms applied into triterpenoids synthesis such as squalene cyclization, hydroxylation, glycosylation, and other structural modifications in which oxidosqualene cyclases (OSC), cytochrome P450 monooxygenases (CYP), and glycosyltransferases (UGTs), which seems to be pivotal in cuticle formation [75]. Our data seems to confirm this observations, thus we have observed significant activation of genes CYP and CER in the fruits of red-fleshed ‘Trinity’ (Figure 8).
Revealed in our research CER1 gene was assigned to the group of CER-like genes previously discovered in different species such as Arabidopsis [79], tomato [80], sweet cherry [81] and orange [82]. Trivedi and coworkers have underlined their skin-specific expression, suggesting that this genes might be responsible for the differential accumulation of very long chain aliphatic compounds [31]. As it was suggested CER genes play important role in alkane biosynthesis, as being linked to aldehyde biosynthesis process and VLCFAs decarbonylation pathway [81, 82]. Another type of genes in fatty acid transformation (such as omega-hydroxypalmitate O-feruloyl transferase) and reductases (like alcohol forming fatty acyl-CoA reductase) and fatty acid hydrolases, generally hampering alcohols production, necessary for fatty acid elongation [76, 52]. CER/FAR interaction, which was firstly determined for Arabidopsis, postulate, that both generate the primary alcohols and alkanes, finally associated with cuticular wax formation [83, 84]. Our results confirm this relation, and highlights this mechanism for the first time in red fleshed apple fruits.
Since the final regulations of phenylopropanoids biosynthesis are energy-intensive, there are several standalone oxidoreductases (CYP) responsible for their molecular transformation, catalyzed in cell structures such as cytochromes and peroxisomes, which up taking the secondary metabolites derived from aromatic amino acids phenylalanine in most plants [85]. As a result of these reactions, different fatty acid conjugates, plant hormones, secondary metabolites, lignin, and many protective chemicals are produced [86–88].
In the membranes of the endoplasmic reticulum, electrons are transferred directly from NADPH to cytochrome via the NADPH-cytP450 reductase complex of flavoprotein, anchored to one layer of the membrane by a hydrophobic chain. For some enzymatic molecules of cytochrome P450, such as cytochrome b5 and cytb5-cytP450 reductase, also a flavoprotein, may participate in the general electron transfer in the flavonoid cycle [89]. The genes CytP450 belonging to the flavoprotein reductases discovered in our study seems to block the flavonoid reductase thus accelerate the anthocyanin biosynthesis. This interaction in red fleshed apple fruits have been explained for the first time.
The crucial enzyme, which functionally catalyst the synthesis of the intermediate 4-(1-methyl-2-pyrrolidinyl)-3-oxobutanoic acid, finally transformed into tropinone through the catalytic activity of the cytochrome P450 is tropinone synthase (CYP82M3) [90].
The biosynthetic pathway of tropinone was fully elucidated in belladonna and it suggests its role in flower petal pigmentation [91]. In our study we have discovered the negative correlation between down regulated TropRed gene (mdm00960), and up regulated Cyp865B1 and CYP86A4 (mdm00073), leading to the activation of alkaloid biosynthesis necessary for final fatty acid elongation (Figure 8.).
Now its sounds:
3.2. Confirmation of contribution of external-pathways genes in anthocyanin biosynthesis
The relation of flavonoids and cutin formation was firstly described in tomato fruits by Heredia and coworkers [56]. They have explained that some of flavonoids are accumulated in cell vacuoles, and others are de novo synthesized and incorporated into the cuticle [56]. This indicates that accumulation of anthocyanins resulting in lower level of cutin formation, following the looseness of apple shelf life ability.
Since the flavanones/anthocyanins and cutin relations have not been yet reported in apples, in this research we have confirmed the complex mechanism of anthocyanin accumulation, demand outside metabolites to be activated.
In this issue, four genes: AlcFARed (alcohol forming fatty acyl-CoA reductase), CER1 (aldehyde decarbonylase), PalmTransf (omega-hydroxypalmitate O-feruloyl transferase) and CYP86A1 (fatty acid hydrolase) mapped on cutin and waxy biosynthesis pathway (KEGG: mdm00073) and three: CYP865B1, CytP450 (probably flavoprotein reductase from cytochrome P450) mapped on peroxisome (KEGG: mdm04146) and TropRed (Tropione reductase) mapped on tropane piperidine and pyridine alkaloid biosynthesis (KEGG: mdm00960) were uncovered to be differentially expressed in red-fleshed fruits of ‘Trinity’.
Apple cuticle is composed of very long-chain fatty acids (VLCFAs, typically between C20–-C34) derived from alkanes, alcohols, esters, aldehydes, ketones and triterpenoids [754, 765]. They are de novo synthesized in plastids under the catalysis of fatty acid (FSA) synthase complex (FAS) generally consisting fatty acyl-ACP thioesterase (FAT) and fatty acid elongase (FAE). In contrast long chain fatty acids are degradeted by acyl-reduction and decarboxylation pathways and mediated by fatty acid transferases [776], and fatty acid hydrolase (CYP86A) (Höfer et al. 2008). Our study underlined the higher activity of AlcFARed, CER1, PalmTransf and CYP86A4 (recognized and FA reductases, transformation of VLCFA, mapped in cutin biosynthesis pathway (KEGG:mdm00073)) in red-fleshed apple fruits and allowed us to uncover the specific relationship between a significant reduction in the amount of VLCF released in fruits and anthocyanin contentspecific relation of significant reduction of VLCFs released in fruits reached in anthocyanins.
A group of molecules employed in cutin forming are triterpenoids, derived from isopentenyl pyrophosphate (IPP, C5). They play a special role in plant cutin layer formation and proceeding with acetyl-CoA (energy precursor in different tissues) [788, 52]. There are also other mechanisms applied into triterpenoids synthesis such as squalene cyclization, hydroxylation, glycosylation, and other structural modifications in which oxidosqualene cyclases (OSC), cytochrome P450 monooxygenases (CYP), and glycosyltransferases (UGTs), which seems to be pivotal in cuticle formation [765]. Our data seems to confirm this observations, thus we have observed noted significant activation of genes CYP and CER in the fruits of red-fleshed ‘Trinity’ (Figure 98).
Revealed in our research CER1 gene was assigned to the group of CER-like genes previously discovered in different species such as Arabidopsis [79], tomato [80], sweet cherry [81] and orange [82]. Trivedi and coworkers have underlined their skin-specific expression, suggesting that this genes might be responsible for the differential accumulation of very long chain aliphatic compounds [31]. As it was suggested CER genes play important role in alkane biosynthesis, as being linked to aldehyde biosynthesis process and VLCFAs decarbonylation pathway [81, 82]. Another type of genes in fatty acid transformation (such as omega-hydroxypalmitate O-feruloyl transferase) and reductases (like alcohol forming fatty acyl-CoA reductase) and fatty acid hydrolases, generally hampering alcohols production, necessary for fatty acid elongation [76, 52]. CER/FAR interaction, which was firstly determined for Arabidopsis, postulate, that both generate the primary alcohols and alkanes, finally associated with cuticular wax formation [83, 84]. Our results confirm this relation, and highlights this mechanism for the first time in red -fleshed apple fruits.
One of the main factor impacting the regulatory mechanism in plant cytochrome and peroxisome is light. This concludes the very intensive connections of those important cell organelles. Since light is essential for plant growth and development some records underline their affection in anthocyanin biosynthesis, (negatively in apple fruits). Generally activated photons inhibits the transcription factors ability [85]. This mechanism have not been investigated yet, and we have retrieved some probable connections between anthocyjanins and wax biosynthesis as well as energy accumulation.
Since the final regulations of phenylopropanoids biosynthesis are energy-intensive, there are several standalone oxidoreductases (CYP) responsible for their molecular transformation, catalyzed in cell structures such as cytochromes and peroxisomes, which up taking the secondary metabolites derived from aromatic amino acids phenylalanine in most plants [895].
As a result of these reactions, different fatty acid conjugates, plant hormones, secondary metabolites, lignin, and many protective chemicals are produced [86–-88].
In the membranes of the endoplasmic reticulum, electrons are transferred directly from NADPH to cytochrome via the NADPH-cytP450 reductase complex of flavoprotein, anchored to one layer of the membrane by a hydrophobic chain.
For some enzymatic molecules of cytochrome P450, such as cytochrome b5 and cytb5-cytP450 reductase, also a flavoprotein, may participate in the general electron transfer in the flavonoid cycle [8990]. The genes CytP450 belonging to the flavoprotein reductases discovered in our study seems to block the flavonoid reductase thus accelerate the anthocyanin biosynthesis. This interaction in red -fleshed apple fruits have been explained for the first time.
The crucial enzyme, which functionally catalyst the synthesis of the intermediate 4-(1-methyl-2-pyrrolidinyl)-3-oxobutanoic acid, finally transformed into tropinone through the catalytic activity of the cytochrome P450 is tropinone synthase (CYP82M3) [901].
The biosynthetic pathway of tropinone was fully elucidated in belladonna and it suggests its role in flower petal pigmentation [912]. In our study we have discovered the negative correlation between down regulated TropRed gene (mdm00960), and up regulated Cyp865B1 and CYP86A4 (mdm00073), leading to the activation of alkaloid biosynthesis necessary for final fatty acid elongation (Figure 98.). In case of red-flashed apple, ,tropinone reductase not fully investigated enzyme because they do not produce tropane alkaloids, thus we have observed negligible its activity in evaluated apple fruit flesh samples (Figure 9.). They seems to be related with apple browning, caused by polyphenol oxidase (PPO) enzymes [91 ]. This is also mechanism that could not be observed in red flesh of apples, that underline the red-fleshed fruits do no browning. However this mechanism must be deeply investigated.
The paragraph of Conclusions was also revised:
According to arrangements made in the manuscript, two reference position were added:
- Liu, H.; Liu, Z.; Wu, Y.; Zheng, L.; Zhang, G. 2021. Regulatory Mechanisms of Anthocyanin Biosynthesis in Apple and Pear. Int. J. Mol. Sci. 22, 8441. https://doi.org/10.3390/ijms22168441
And
- Birgit Dräger B. 2006. Tropinone reductases, enzymes at the branch point of tropane alkaloid metabolism, Phytochemistry, 67, 327-337; https://doi.org/10.1016/j.phytochem.2005.12.001
- Additionally, the paper would be stronger with a more explicitly formulated conclusion. The current ending summarizes the results but does not clearly state what the authors see as the next steps or main implications of their findings. The authors must improve this section by explaining how their discoveries might contribute to future breeding strategies, storage quality improvement, or the development of functional fruit varieties. Defining a clear direction for future breeding research and underlining the societal or economic importance of this work would give the article a more complete and purposeful closing.
Thak You for pointing it out, the coclusions were to general and indeed did not geve any practical summary of our work.
The remark has been accepted.
The concusions now summarise and outlining the practical and commercial and applicable significance, as well as the role of red-fleshed apples in breeding programs.
It was:
- Conclusions
Our analysis allowed us to discover new insights into the regulation of apple flesh color acquisition. We found an undoubted link between the anthocyanidin biosynthetic pathways and parallel metabolism involved fruit wax coating, controlled by factors, such as: N3Dioxy (naringenin 3-dioxygenase), LAR1 (leucoanthocyanidin reductase 1) and F3Mo (flavonoid 3-monooxygenase) mapped to the flavonoid synthesis pathway (KEGG: mdm00941); AlcFARed (alcohol-forming fatty acyl-CoA reductase), CER1 (aldehyde decarbonylase), PalmTransf (omega-hydroxypalmitate O-feruloyl transferase) and CYP86A1 (fatty acid hydrolase) - mapped to the cutin and wax biosynthetic pathway (KEGG: mdm00073); and CYP865B1, CytP450 (probably a cytochrome P450 flavoprotein reductase) mapped to peroxisomes (KEGG: mdm04146); and TropRed (tropione reductase) mapped to tropane piperidine and pyridine alkaloid biosynthesis (KEGG: mdm00960).
All discovered genes showed significant correlations with total anthocyanin content and visible fruit flesh color in the fruit flesh of analyzed genotypes. Although, many reports exist on anthocyanin development, both in the peel and fruit flesh, the complexity of this phenomenon is not fully understood. We confirmed that the interaction of the flavonoid biosynthetic pathway and the fatty acid degradation mechanism may overlap during fruit formation. However, the final postulates has to be validated, especially in the context of introduction of selected sequences as potential candidate genes for conducting MAS procedure in future breeding programs of red-fleshed apple trees.
In the final version, the conclusions paragraph sound like:
Our analysis revealed novel molecular interactions between anthocyanin biosynthesis and wax metabolism pathways in apple fruit flesh. The identification of ten key genes—N3Dioxy, LAR1, F3Mo (flavonoid pathway); AlcFARed, CER1, PalmTransf, CYP86A4 (wax biosynthesis); and TropRed, CyP865B1, CytP450 (alkaloid biosynthesis/peroxisome)—provides new insights into the molecular basis of flesh coloration and storage stability relationships.
The identified genes offer immediate potential as molecular markers for marker-assisted selection (MAS) in red-fleshed apple breeding programs. Priority should be given to developing functional markers for: CER1 and AlcFARed expression levels as predictors of storage stability, F3Mo/LAR1 ratio as an indicator of flesh color intensity. Combined marker assay for simultaneous selection of color and storage traits.
Our findings suggest that red-fleshed varieties' storage limitations stem from reduced wax biosynthesis capacity. Future research should focus on developing cultivation practices to enhance CER1 and CYP86A4 expression, investigating postharvest treatments targeting wax metabolism pathways, and finally exploring genetic modification approaches to restore wax biosynthesis in high-anthocyanin backgrounds
Successfully addressing storage limitations could unlock significant market potential for red-fleshed apples, within functional food sectors. Improved varieties would support sustainable agriculture by providing growers with premium products while delivering enhanced nutritional benefits to consumers.
Future investigations should validate these gene-trait relationships across diverse genetic backgrounds, develop high-throughput screening methods for breeding programs, and explore environmental factors influencing the anthocyanin-wax metabolism balance. Integration with genomics-assisted breeding platforms will be essential for translating these molecular insights into commercial cultivar development.
- There are, moreover, a few minor issues that could be improved. Some parts of the text would benefit from clearer and more concise English phrasing. For example, in the abstract, the sentence “Our results, postulate, that the fatty acid degradation process is initiating in flesh of apple fruits…” should be corrected to “Our results postulate that the fatty acid degradation process is initiated in the flesh of apple fruits.”
Many thanks for the suggestion.
The remark has been accepted, and some English phrasing were corrected.
- In addition, the cultivar name “Free Redstar” is occasionally misspelled as “Free Redsar,” which should be unified.
Thank You for pointing this out. The remark has been accepted. Authors made proper corrections. The term was unified in entire document.
- The use of “red-fleshed” versus “red fleshed” is also inconsistent and should be standardized throughout the manuscript.
Thank You for pointing this out. The remark has been accepted. Authors made proper corrections. The term was unified in entire document.
- Comments on the Quality of English Language
Thank You for all valuable suggestion. The manuscript was revised by our English expert.

Round 2
Reviewer 1 Report
Comments and Suggestions for Authors
In this version of the manuscript, the introduction and discussion sections have not been logically revised. The concept map lacks clarity and contains weak logic. Figure 1 has issues with significance labeling, which needs to be carefully checked. Besides, it is necessary to check the compliance of the figures and tables in the main text.
Comments on the Quality of English LanguagePay attentions to grammars, academic conventions, consistency and flow of the manuscript.
Author Response
7.11.2025
Keller-Przybylkowicz Sylwia
The National Institute of Horticultural Research;
Konstytucji 3-go Maja, 96-100 Skierniewice, Poland
International Journal of Molecular Science
Dear Managing Editor
Dear Lvy Ni
[Cover Letter]
We would like to thank reviewer for every remark that helped us to improve the current version of the manuscript.
We appreciate you for your precious time in proofing our paper and providing valuable comments.
We have carefully considered the comments from the second round of review and accepted the suggestions. We hope the final version of the manuscript, prepared after careful revisions meet your high standards.
All modifications in the manuscript have comments in current version accordingly to the track changes of word document and in fixed pdf.
Sincerely,
Sylwia Keller-Przybylkowicz,
Sylwia.Keller@inhort.pl
The National Institute of Horticultural Research
Konstytucji 3-go Maja 1/3
96-100 Skierniewice
Poland
Author’s responses to the comments of the reviewer #1 of the manuscript entitled “Molecular interaction of genes related with anthocyanin, lipid and wax biosynthesis in apple red-fleshed fruits., submitted to International Journal of Molecular Science under the tracking number: ijms-3954842.
The authors would like to thank the reviewer for valuable comments. We acknowledge that the first revised version of the manuscript did not maintain proper flow, and we appologise for this misundertanding. We absolutely agree with this valuable suggestion. This likely resulted from a misunderstanding of the reviewer's too general comments.
Reviewer remark:
In this version of the manuscript, the introduction and discussion sections have not been logically revised. The concept map lacks clarity and contains weak logic.
Pay attentions to grammars, academic conventions, consistency and flow of the manuscript.
Remark has been acceted
The manuscript has been revised to maintain proper information flow. Some paragraphs have been moved from the discussion section to the introduction, and vice versa.
This rereangments were done based on the egxisting basic text.
To maintain logical continuity, a few sentences have also been added. All document reorganizations are visible in the change tracking file.
For this purpose some senteses were added:
Introduction:
Apples is are one of the most important fruit crops worldwide. The gGlobal annual apple annual production haves reached over 80 millionln tons, and about 45% of its production is covered byoccurs in China, followed by Turkey and the USA (FAOSTAT 2024). Such aThe massive supply of this fruits to the market and its management raises challenges in terms of cultivation and storage. An One opportunity way to address this situation lies ininvolves increasing the consumption of fresh apples and their products (such as juices, chips and ciders), which can be achieved by introducing red-fleshed varieties into the apple industry red-fleshed varieties [1].
Results:
2.1. Significant differences of in apple genotypes with regard to total anthocyanin content
The calculated average amount of total anthocyanin content ranged from 9,482 mg/100ml (in ‘Free Redstar’, control) to 321 mg/100ml (in genotype no. 48) (Table 1). The most informative and variable significance between evaluated samples is presented in Figure 1.
The table 1 was added:
Table 1. The average anthocyanin concentrations measured for 10 apple fruits collected from each genotype. The standard error (SE) was calculated in accordance to compartment whit ‘Free Redstar’
|
Cultivar |
Average anthocyanin concentration |
SE difference |
|
Free Redstar |
9,482 |
- |
|
Trinity |
297,472 |
3,078 |
|
48 |
321,403 |
6,681 |
|
154 |
290,723 |
4,779 |
|
44 |
184,556 |
5,070 |
|
40 |
241,094 |
5,647 |
|
84 |
164,864 |
4,643 |
|
103 |
77,586 |
2,197 |
|
141 |
24,403 |
0,9361 |
|
72 |
20,797 |
1,173 |
|
77 |
29,968 |
1,801 |
|
126 |
12,489 |
1,081 |
Discussion;
In general, this initialted analysis confirmed, the activity of the structural genes from the anthocyanin biosynthesis pathway, as well as the basic mechanisms in of red-fleshed fruit, ongoing influencingin the evaluated apple hybrid genotypes.
Interestedly, in our research, we have discovered ten genes potentially bridging the anthocyanin metabolism with other external pathways, involved in the stimulation of their biosynthesis in apple fruit flesh. Those genes, mapped on through the flavonoid biosynthesis, cutin/suberin/wax biosynthesis, tropane/piperidine/pyridine alkaloid biosynthesis and peroxisome pathways and peroxisome (Figure 9), have not been investigated so far.
Due to the rearrangement of the document's paragraphs, the References section has been reorganized according to the IJMS standard.
The quality of the figures has been improved. Corrections were made to the best of the authors' ability.
A full English proofreading was performed for a fee at the request of MDPI (AuthorSevis).
According to the englich editor suggestion the title of the mauscript has been edited and now is:
„Molecular interaction of genes related to anthocyanin, lipid and wax biosynthesis in apple red-fleshed fruits”
Previously was: „Molecular interaction of genes related with anthocyanin, lipid and wax biosynthesis in apple red-fleshed fruits.”
We belive that this final version of the manuscript wil be acceptible to published, and meets the scientific standards of the joiurnal.
Reveiwer remark
Figure 1 has issues with significance labeling, which needs to be carefully checked.
Remark has been accepted
Thank You for the suggestio. Authors used the specyfic softwar to prepare the diagram.
Since the differences between the samples in accordancre to te anthocyanin concentration (with the significans ****p≤0,0001) are significantly high, authors decited to show the sample-to-sample relations with more variable significance. To clarify presented context, we decited to add propare table to show the relative avarage amount of anthocyjanin measuerments performer for ech genetype studied. Therefore, the divergence of significance informations were highlited in the bar plot.
We hope this revisions meets the reviewer requirements.
Reviewer remark:
Besides, it is necessary to check the compliance of the figures and tables in the main text.
Many thanks for remark. The compliance of the figures and tables were checked.

Round 3
Reviewer 1 Report
Comments and Suggestions for Authors
I have no further comments at this time.
Comments on the Quality of English LanguagePay attentions to grammars, academic conventions, consistency and flow of the manuscript.